# T cells specific for post-translational modifications escape intrathymic tolerance induction

Bruno Raposo[1,8], Patrick Merky[1], Christina Lundqvist[3], Hisakata Yamada[2], Vilma Urbonaviciute[1], Colin Niaudet[4], Johan Viljanen[5], Jan Kihlberg[5], Bruno Kyewski[6], Olov Ekwall [3,7], Rikard Holmdahl [1] & Johan Bäcklund[1]

Establishing effective central tolerance requires the promiscuous expression of tissue-restricted antigens by medullary thymic epithelial cells. However, whether central tolerance also extends to post-translationally modified proteins is not clear. Here we show a mouse model of autoimmunity in which disease development is dependent on post-translational modification (PTM) of the tissue-restricted self-antigen collagen type II. T cells specific for the non-modified antigen undergo efficient central tolerance. By contrast, PTM-reactive T cells escape thymic selection, though the PTM variant constitutes the dominant form in the periphery. This finding implies that the PTM protein is absent in the thymus, or present at concentrations insufficient to induce negative selection of developing thymocytes and explains the lower level of tolerance induction against the PTM antigen. As the majority of self-antigens are post-translationally modified, these data raise the possibility that T cells specific for other self-antigens naturally subjected to PTM may escape central tolerance induction by a similar mechanism.

[1] Section for Medical Inflammation Research, Department of Medical Biochemistry and Biophysics, Karolinska Institutet, Scheeles väg 2, 17177 Stockholm, Sweden. [2] Division of Host Defense, Medical Institute of Bioregulation, Kyushu University, Fukuoka 812-8582, Japan. [3] Department of Rheumatology and Inflammation Research, Institute of Medicine, The Sahlgrenska Academy, University of Gothenburg, 41346 Gothenburg, Sweden. [4] Division of Vascular Biology, Department of Medical Biochemistry and Biophysics, Karolinska Institutet, 17177 Stockholm, Sweden. [5] Department of Chemistry, BMC Uppsala University, 751 23 Uppsala, Sweden. [6] Division of Developmental Immunology Tumor Immunology Program, German Cancer Research Center (DKFZ), 69120 Heidelberg, Germany. [7] Department of Pediatrics, Institute of Clinical Sciences, the Sahlgrenska Academy, University of Gothenburg, 41346 Gothenburg Sweden. [8] Present address: Department of Microbiology and Immunobiology, Harvard Medical School, 77 Avenue Louis Pasteur, NRB 836, Boston, MA 02115, USA. Rikard Holmdahl and Johan Bäcklund jointly supervised this work. Correspondence and requests for materials should be addressed to R.H. (email: Rikard.Holmdahl@ki.se)

Central T cell tolerance is established in the thymus where developing thymocytes that react strongly with self-antigens are either negatively selected and depleted or alternatively deviated into the T regulatory cell lineage. Central tolerance not only encompasses ubiquitous and circulating self-antigens, but also a large set of tissue-restricted self-antigens (TRAs) that are ectopically expressed in medullary thymic epithelial cells (mTEC)[1,2]. The expression of TRA by mTEC is to a large extent controlled by the autoimmune regulator (Aire) protein, and dysfunction of Aire is associated with defective central tolerance and autoimmune disorders[3–5]. Nevertheless, auto-reactive T cells exist in the periphery of healthy individuals, indicating that central tolerance is incomplete[1,2,6].

Most proteins are subject to different types of post-translational modifications (PTMs), e.g., phosphorylation or glycosylation, which often change the structure and function of the protein. Moreover, PTM are also likely to change the way the protein is processed and recognized as a self-antigen by immune cells. T cell reactivity to PTM self-antigens has been considered to be an initiating and/or perpetuating factor in the progression of autoimmune diseases. PTM of self-antigens have been reported in

patients with rheumatoid arthritis (RA)[3–5,7] and type 1 diabetes[8]. Such modifications have been shown to affect the binding of the antigen to the major histocompatibility complex (MHC) molecule, and consequently affect T cell activation[7,8]. A similar mechanism has been described in detail for celiac disease, in which modification of gliadin peptides enable recognition by gut T cells[9]. PTM can occur spontaneously, like oxidation or nitrosylation, or be enzyme-mediated like citrullination and glycosylation. In either case, the occurrence and degree of PTM is dependent on a number of host factors such as the compartmentalization of the enzyme or protein, regions flanking the amino acid to be modified, as well as physiological factors like pH and redox states. PTM of a self-antigen may occur naturally, in order to generate the desired biological activity of a protein, or in response to infection, inflammation, or physical damage. In the latter scenarios, creation or exposure of neo-epitopes to which the immune system has not been previously tolerized is possible. Nevertheless, leaky self-tolerance might also arise under non-pathological conditions. In this situation, the PTM self-antigen would only occur in peripheral tissues, while being absent from central lymphoid organs such as the thymus.

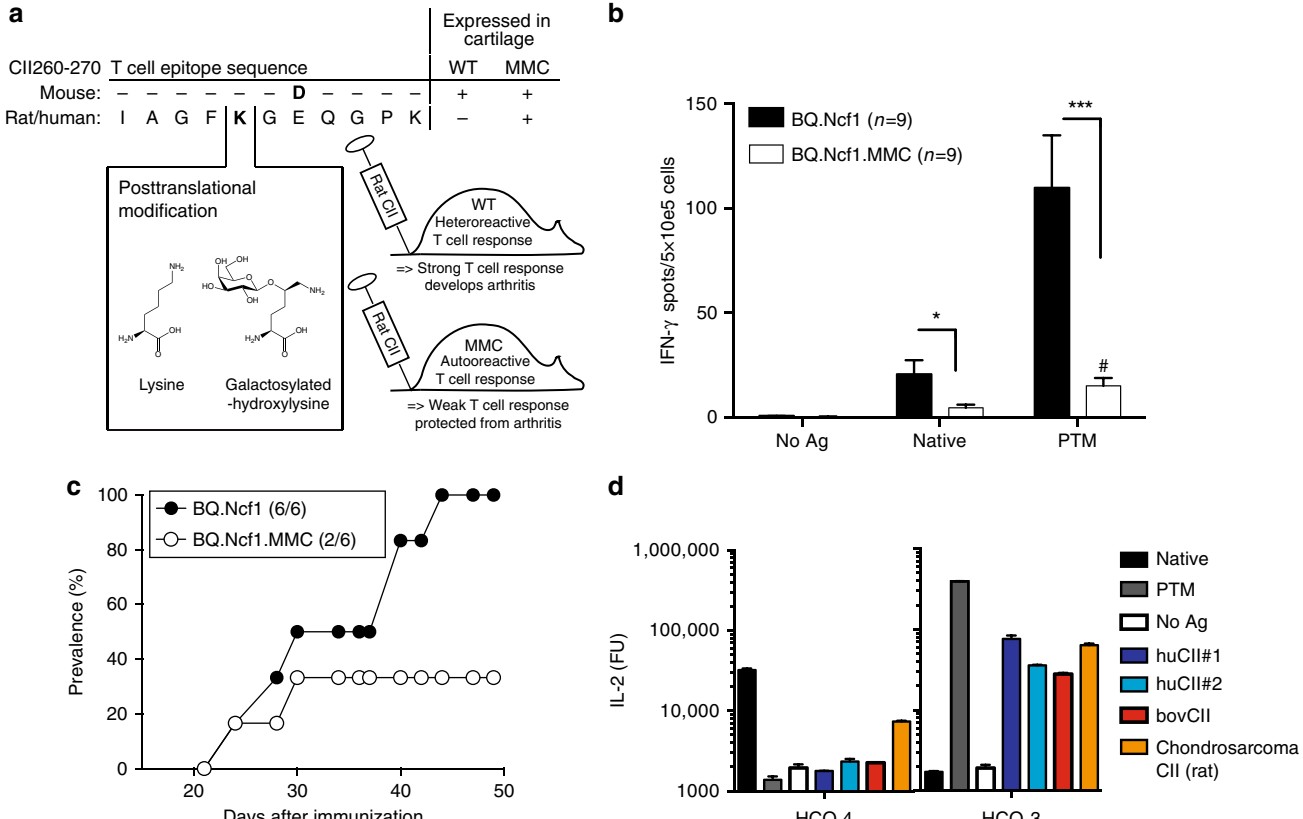

**Fig. 1** Summary of the autologous CIA model. **a** The MMC mouse expresses the immunodominant T cell epitope of heterologous (rat/human) CII as a self-antigen in the cartilage. A D266E amino acid substitution is the only difference between mouse and heterologous CII. A lysine at position 264 can become post-translationally modified by hydroxylation (not shown) and subsequent glycosylation with a monosaccharide. These modifications are recognized by distinct T cell clones (shown in **d**). MMC mice are immunized with heterologous CII (in this study rat CII) in order to induce an autoreactive T cell response. **b** The $Ncf1$ mutation was inserted onto the BQ background in order to render arthritis susceptibility in MMC mice. IFN-$\gamma$ ELISPOT of lymph node cells isolated 10 days after immunization with rat CII and following re-stimulation with the non-modified (native) or glycosylated (PTM) CII peptide are shown. Bars indicate the mean number of spots ± SEM. #Statistical significance ($p = 0.0195$) between native and PTM responses in BQ.Ncf1.MMC mice, using a Wilcoxon's test. **c** Arthritis susceptibility of MMC mice in the BQ.Ncf1 background is reduced but not completely abrogated, when compared to non-MMC littermates. Number in brackets indicates cumulative number of animals that developed arthritis over total number of animals. **d** IL-2 production of T cell hybridoma clones specific to either the non-modified (HCQ.4) or PTM (HCQ.3) variants of the CII$_{260-270}$ epitope after stimulation with the different peptides as well as whole CII protein obtained from different species (hu, human (two independent preparations/samples); bov, bovine (calf) CII from joint cartilage. The chondrosarcoma is a rat tumor line that was used as a positive control as it produces CII that is very heterogeneous in terms of PTMs[19]). Mann–Whitney $U$ test was used in ELISPOT assays. *$p < 0.05$; ***$p < 0.001$

With regard to central tolerance against TRA, it is unknown whether antigen-presenting cells (APC) in the thymus have all the necessary machinery and/or whether the intracellular environment allows for all potential PTM to be generated and displayed in order to impose tolerance. There are many examples of autoreactive T cell responses specifically directed against PTM variants of self-antigens[10–12]; however, a formal demonstration that PTM are exempt from central tolerance is required.

## Results

**Aire mediates tolerance to the native antigen and not PTMs.** In order to investigate how physiological PTM of autoantigens may affect central tolerance and the susceptibility to a tissue-restricted autoimmune disease, we made use of the autologous collagen-induced arthritis (CIA) model for RA. In this model, mice expressing a point-mutated collagen type II (CII) molecule mimicking the human/bovine/rat T cell epitope (MMC mouse, for mutated mouse collagen[13]; Fig. 1a) can be immunized with either of these CII molecules. Due to accessibility, we have used rat CII in our immunization protocols. Whereas in the traditional CIA model the T cell response is raised solely against the immunized foreign CII protein, with no cross-reactivity to mouse self-CII; in the autologous CIA model, the T cell responses are directed against the heterologous CII expressed in the joint cartilage of the MMC mouse. Hence, efficient tolerization of T cells specific for the immunodominant CII epitope present on human, rat, bovine, or chicken CII (amino acids 260–270; $CII_{260-270}$) can only take place in the MMC mouse[14]. Importantly, expression of the heterologous CII molecule (in MMC mice) has been shown to be naturally regulated and to occur exclusively at sites of known physiological CII expression, e.g., joint cartilage and the eye[13].

The tolerance status in MMC mice can be determined by analyzing the ex vivo phenotype and activation of $CII_{260-270}$ specific T cells, and by monitoring the susceptibility of MMC mice to develop autoimmune arthritis. Most importantly, the $CII_{260-270}$ epitope is naturally subjected to PTM. The lysine at position 264 constitutes a critical TCR contact point and is naturally subjected to hydroxylation and glycosylation, where each of the two variants is recognized by distinct T cell clones[15,16]. To induce autoimmunity against self-CII, MMC mice and control wild-type littermates (WT) were immunized with rat CII and monitored for development of arthritis. All mice had a mutation in the *Ncf1* gene, which enhances arthritis susceptibility and allows for the development of arthritis in MMC mice[17]. The

expression of heterologous CII in the joints of MMC mice resulted in fewer numbers of activated CII-reactive T cells, after antigen immunization (Fig. 1b). Consequently, MMC mice were less susceptible to arthritis (Fig. 1c). These observations suggest that T cells in the MMC mouse are tolerized to heterologous CII. Nevertheless, such tolerance remained incomplete. Whereas an almost completely abrogated response was observed against the native form of $CII_{260-270}$, T cell reactivity to the PTM $CII_{260-270}$ peptide remained significant ($p = 0.0195$, Wilcoxon's test)[18]. The absence of response to the non-modified $CII_{260-270}$ epitope suggests the possibility that the native variant is abundantly available for tolerance induction in vivo. To test this, we extracted CII from healthy joint cartilage of both human and bovine sources, and evaluated their level of PTMs. In both cases, extracted CII was only able to activate T cell clones specific for the PTM variant (Fig. 1d), supporting an earlier observation that CII from healthy cartilage is uniformly glycosylated in both rodents and humans[19].

Based on these observations, we note an apparent discrepancy in the relative abundance of CII expression in the periphery and the overall level of tolerance. Hence, the data suggest that tolerance towards the two variants of the antigen may be differentially regulated. In order to investigate a potential role of the thymus in this biased tolerance regulation, we generated Aire-sufficient and Aire-deficient MMC mice (hereafter denoted as $MMC.Aire^{Suf}$ and $MMC.Aire^{KO}$, respectively). Arthritis protection in the autologous CIA model was found to be *Aire*-dependent, as $MMC.Aire^{KO}$ mice developed more arthritis, and a stronger humoral response, when compared to $MMC.Aire^{Suf}$ littermates (Fig. 2a and Supplementary Fig. 1a and b).

Analyses of T cell recall responses early after immunization, or during the chronic phase of disease, showed that the higher arthritis susceptibility of $MMC.Aire^{KO}$ mice was coupled to a specific increase in reactivity against the non-modified antigen (Fig. 2b, c). Whereas $MMC.Aire^{Suf}$ mice mounted a substantial response only to the PTM peptide, $MMC.Aire^{KO}$ mice displayed an immunodominant response to the non-modified peptide, with an unaltered PTM peptide response. Similar data were also obtained for IL-17A-producing cells (Supplementary Fig. 2b).

**CII is promiscuously expressed in mouse and human mTECs.** Taken together, these observations strongly suggest that tolerance to self-CII is dependent on its molecular structure (native or PTM form), and that it is controlled at the thymic level. The Aire

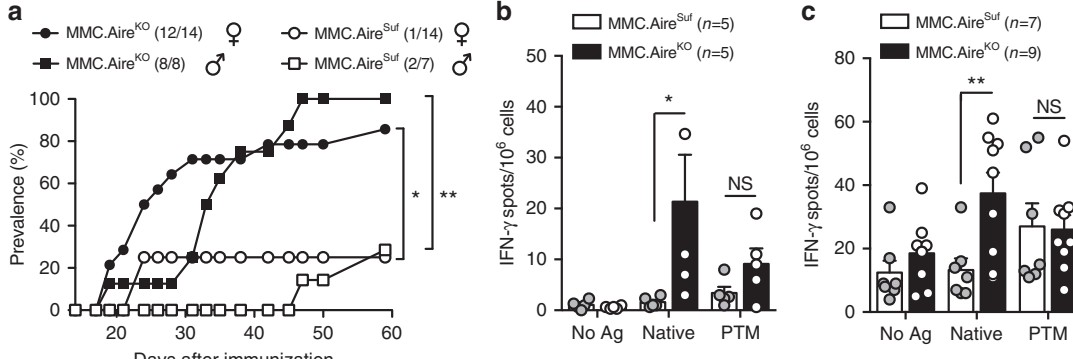

**Fig. 2** Tolerance to the non-modified $CII_{260-270}$ epitope is lost after Aire deficiency. Loss of tolerance to CII occurs in an Aire-dependent way and specifically affects T cells specific for the non-modified $CII_{260-270}$ epitope. **a** Arthritis prevalence in $MMC.Aire^{Suf}$ and $MMC.Aire^{KO}$ male and female mice after immunization with rat CII. **b**, **c** ELISPOT data from in vitro recall responses of pooled lymph node and spleen cells from indicated number of MMC. $Aire^{Suf}$ and $MMC.Aire^{KO}$ mice 10 days (**b**) or 10 weeks (**c**) after CII immunization. Cells were stimulated with the non-modified and PTM $CII_{260-270}$ peptide or left unstimulated (No Ag). Values shown are the mean ± SEM number of spots recorded for IFN-γ producing cells. Fisher's exact test was used to calculate significance of arthritis prevalence, whereas Mann–Whitney $U$ test was used in ELISPOT assays. *$p < 0.05$; **$p < 0.01$; NS not significant

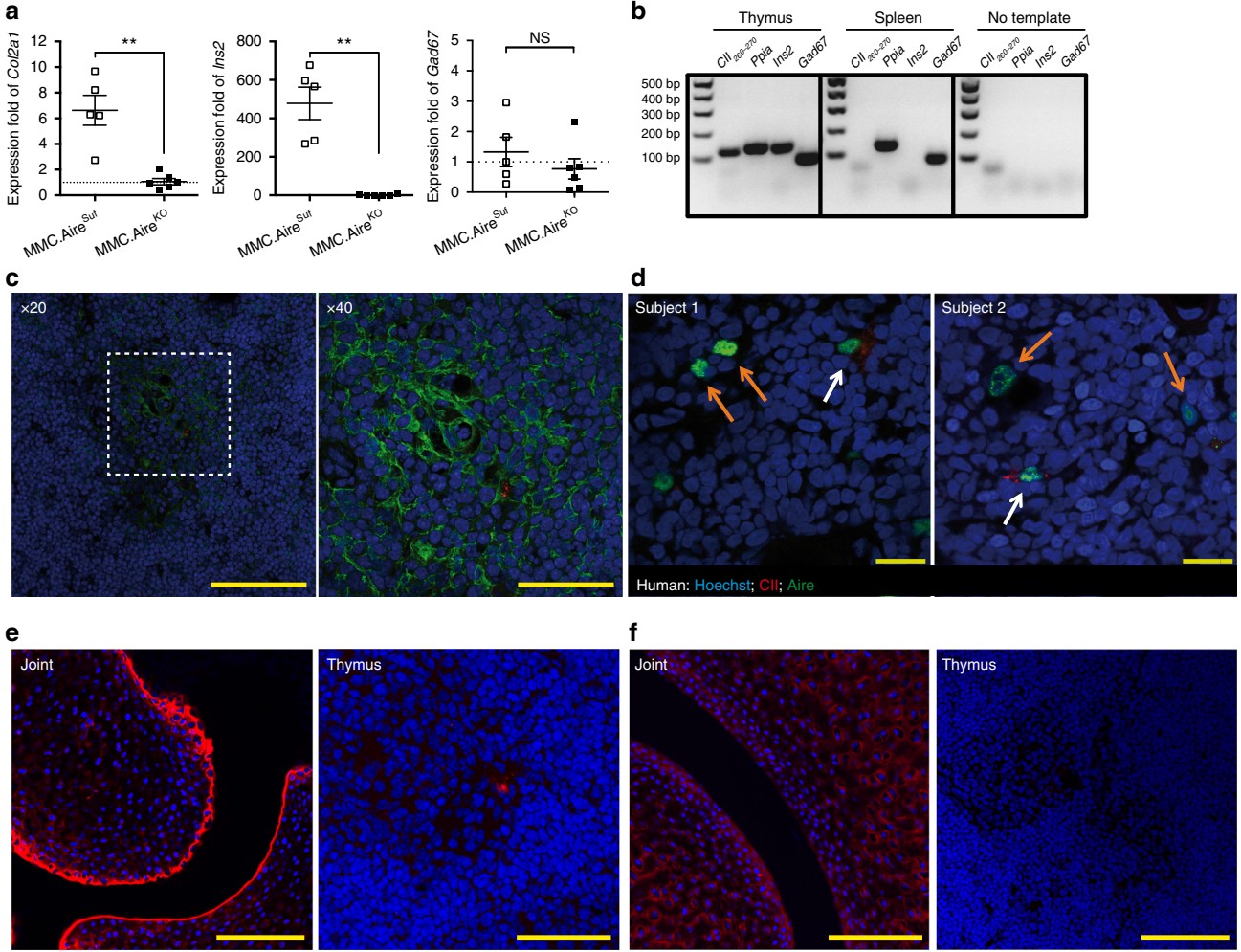

**Fig. 3** The CII epitope is expressed in thymic stromal cells of mice and humans. **a** cDNA derived from thymi of MMC.Aire$^{Suf}$ ($n = 5$) and MMC.Aire$^{KO}$ ($n = 6$) was prepared and expression of CII (*Col2a1*) was determined by quantitative RT-PCR. Aire-dependent (*Ins2*) and Aire-independent (*Gad67*) genes were used as controls. Data were normalized to the expression of cyclophilin A (*Ppia*) and calibrated with one MMC.Aire$^{KO}$ sample. Mean ± SEM is shown. **b** Qualitative expression analysis of the immunodominant T cell epitope CII$_{260-270}$ by RT-PCR on cDNA prepared from whole thymi of 3-week-old mice. As negative control, cDNA was prepared from spleen cells of wild-type mice. *Ins2* and *Gad67* were used as controls for genes expressed either in the thymus alone or in both thymus and spleen, respectively. *Ppia* was used as housekeeping gene, whereas no template samples were used as negative controls. **c** MMC thymus stained for DNA (Hoechst, in blue), keratin 5 (in green), and CII (mAb cocktail, in red). Scale bar in left panel, 100 μm; in right panel, 50 μm. **d** Human thymus sections from two different subjects stained with Hoechst (in blue), anti-AIRE (in green), and anti-CII (in red). Orange arrows indicate thymic epithelial cells positive for AIRE alone. White arrows indicate thymic epithelial cell positive for both AIRE and CII. Scale bar indicates 20 μm. (**e**) Stain of joint and thymus with Hoechst (blue) and anti-CII (mAb cocktail, red) or (**f**) Hoechst (blue) and anti-PTM CII (T8 mAb, red), from an MMC mouse. Scale bars indicate 100 μm for left panel of **e** and both panels of **f**, and 50 μm in right panel of **e**. *p* values were calculated using Mann–Whitney U test. **\*\***$p < 0.01$; NS not significant

dependence of thymic CII expression has been controversial[20,21]. The discrepancies may, in part, be explained by different mouse strains and normalization of gene expression data[22]. Using insulin (*Ins2*) and glutamic acid decarboxylase 67 (*Gad67*) as Aire-dependently and Aire-independently expressed prototypic genes, respectively, we found CII (*Col2a1*) to be expressed in the thymus in a partially Aire-dependent manner (Fig. 3a, b; and Supplementary Fig. 3 for WT mice). CII mRNA transcripts were only detected in the thymus but not in secondary lymphoid organs (e.g., spleen), and included the CII$_{260-270}$ epitope, as determined by reverse transcription-polymerase chain reaction (RT-PCR) (Fig. 3b) and Sanger sequencing. Using monoclonal antibodies specific for triple helical CII[23,24], we were able to confirm expression at the protein level in murine and human thymus samples (Fig. 3c, d, respectively). Interestingly, CII$^+$ cells coincided with medullary regions and mTEC (Fig. 3d and

Supplementary Fig. 4). However, contrary to CII in the joints, the K264 on CII was not glycosylated when CII was expressed in the thymus (Fig. 3e, f). Quantification of CII$^+$ cells in murine and human thymi (Supplementary Table 1) suggests, once again, that CII constitutes a self-antigen involved in central selection of thymocytes.

**Central tolerance is induced by presentation of native epitope.** The specificity of the T cell receptor is promiscuous[25]. To exclude the possibility that thymic selection towards the non-modified CII$_{260-270}$ peptide (Figs. 1b and 2b, c) did not result from thymic presentation of an unrelated (but Aire-dependent) autoantigen to which these T cells would cross-react, we grafted 3–4 weeks old naturally athymic nude mice (H-2A$^q$) with MMC or WT thymi. After establishment of a peripheral T cell pool (>95% of recipient

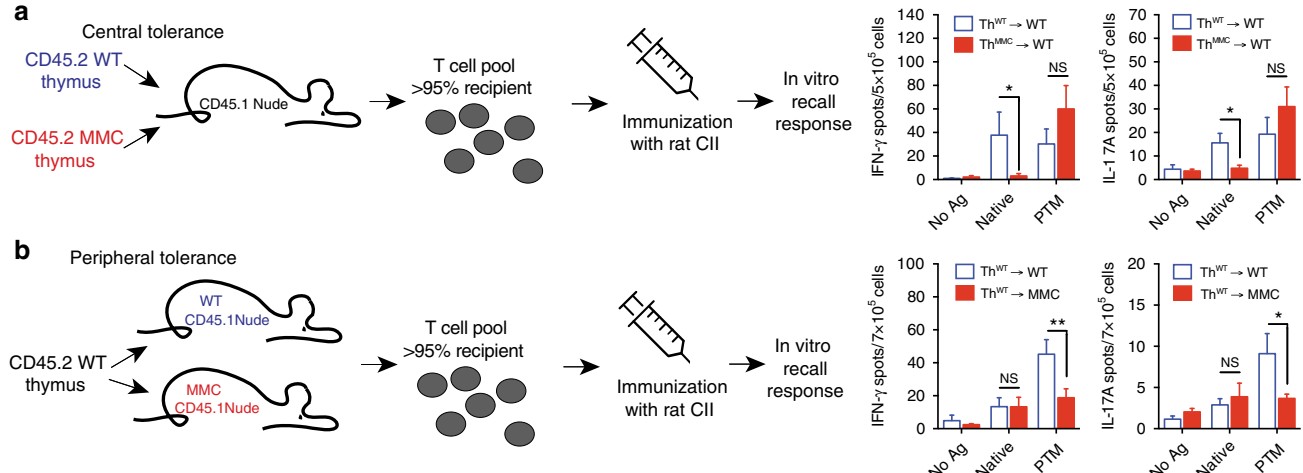

**Fig. 4** mTEC-mediated central tolerance to self-CII is limited to the native epitope. **a** Four-week-old nude mice were grafted with neonate thymi from either WT or MMC donors (n = 5/group). Eleven weeks later, mice were immunized with rat CII and draining lymph node cells were re-stimulated in vitro with different CII peptides. **b** MMC-positive (n = 8) and MMC-negative (n = 6) nude mice were grafted with neonate thymi from wild-type donors. Eleven weeks later, recipients were immunized with rat CII and draining lymph node cells were re-stimulated in vitro with different CII peptides. Values shown are the mean ± SEM number of spots recorded for IFN-γ-producing and IL-17A-producing cells. p values were calculated using Mann–Whitney U test. *p < 0.05; **p < 0.01; NS not significant

origin; Supplementary Fig. 5), recall responses to CII were determined. In this scenario, imprinting of the T cell repertoire of naive thymocytes by the heterologous $CII_{260-270}$ epitope can only be mediated via thymic stromal cells in recipients grafted with a MMC thymus. In agreement with the data shown in Fig. 1, nude mice grafted with an MMC thymus displayed a significantly reduced T cell response to the non-modified $CII_{260-270}$ peptide, compared to nude mice grafted with a WT thymus (Fig. 4a). Most importantly, the recall response towards the PTM $CII_{260-270}$ peptide remained identical, regardless of the donor thymus. Since the only difference between MMC and WT donors is a single amino acid substitution at position 266 within the $CII_{260-270}$ epitope, the reduced recall response in MMC-thymus recipients must have resulted from tolerance induced through presentation of the $CII_{260-270}$ epitope and not another unrelated self-antigen.

The PTM $CII_{260-270}$ variant is dominantly expressed in healthy cartilage (Fig. 1d and ref. [19]). Nevertheless, the non-modified peptide may also be available in the periphery for immune recognition and tolerance induction. Therefore, the simultaneous existence of both central and peripheral tolerance mechanisms may explain the biased tolerance towards the non-modified $CII_{260-270}$ epitope. To address this, we established MMC-expressing nude mice, grafted them with WT thymus, and determined recall responses after reconstitution (Fig. 4b). MMC-nude mice displayed a strongly reduced recall response to the PTM peptide. However, the recall response to the non-modified peptide remained unaltered, compared to controls (WT nude mice grafted with WT thymi). Taken together, the data suggest that tolerance to the native $CII_{260-270}$ is established at the thymic level, whereas peripheral mechanisms appear to play no or only a very minor role in establishing tolerance to non-modified CII. Conversely, tolerance to the PTM peptide observed in MMC-nude recipients can conceivably only originate from the peripheral expression.

**Migratory APCs can transfer PTM CII to the thymus.** To investigate to what extent the thymus contributes to central tolerance to the PTM epitope variant, we developed a TCR transgenic mouse line expressing an αβTCR (HCQ3)[15], which is

specific for the PTM variant of the heterologous $CII_{260-270}$ peptide in the context of $A^q$. Importantly, transgenic HCQ3 T cells do not cross-react with the non-modified CII peptide (Supplementary Fig. 6a). When transferred to naive WT or MMC recipients, naive HCQ3 T cells proliferated vigorously in joint-draining lymph nodes of MMC, but not WT mice, as assessed by carboxyfluorescein succinimidyl ester (CFSE) dilution (Supplementary Fig. 6b). Furthermore, HCQ3.MMC double-positive mice displayed a reduced recall response to the PTM peptide (Supplementary Fig. 6c). Together, these data show that the PTM antigen is indeed displayed for immune recognition in the periphery of naive MMC mice. Moreover, it shows that HCQ3 T cells are only tolerized when in the context of MMC co-expression.

Next, we determined the frequency of thymocytes specific for the PTM peptide in HCQ3.MMC compared with single-positive HCQ3 littermates. There was a minor but significant decrease of CD4 single-positive (CD4SP) thymocytes in HCQ3.MMC mice (Fig. 5a). Due to lack of reliable peptide–MHC class II tetramers for $CII_{260-270}$, we assessed the immediate up-regulation of CD40 ligand (CD40L) on stimulated T cells. CD40L has previously been shown to serve as a specific and unbiased marker for antigen-specific T cells regardless of their cytokine profile[26]. Upon ex vivo stimulation with the PTM peptide, we observed that the frequency of HCQ3.MMC thymocytes able to up-regulate CD40L was significantly reduced, in comparison to HCQ3 thymocytes (Fig. 5b).

Thymic selection by Aire-expressing mTEC depends on the interaction between specific TNF family members, such as RANK-RANKL and CD40-CD40L[27–30]. Mice treated with anti-RANKL monoclonal antibody (mAb) are transiently depleted of Aire-expressing mTEC, which consequently affects negative selection[31]. We then treated HCQ3.MMC mice with anti-RANKL mAb and assessed T cell selection. Indeed, Aire-expressing mTEC were depleted after mAb treatment (Fig. 5c). Nevertheless, considering the similar levels of HCQ3 cells able to respond to the PTM peptide, ablation of Aire+ mTECs did not affect negative selection of HCQ3 thymocytes (Fig. 5d). Taking into consideration the data shown in Figs. 2 and 4, this observation supports the notion that tolerance to the non-modified antigen is established in the thymus and is dependent

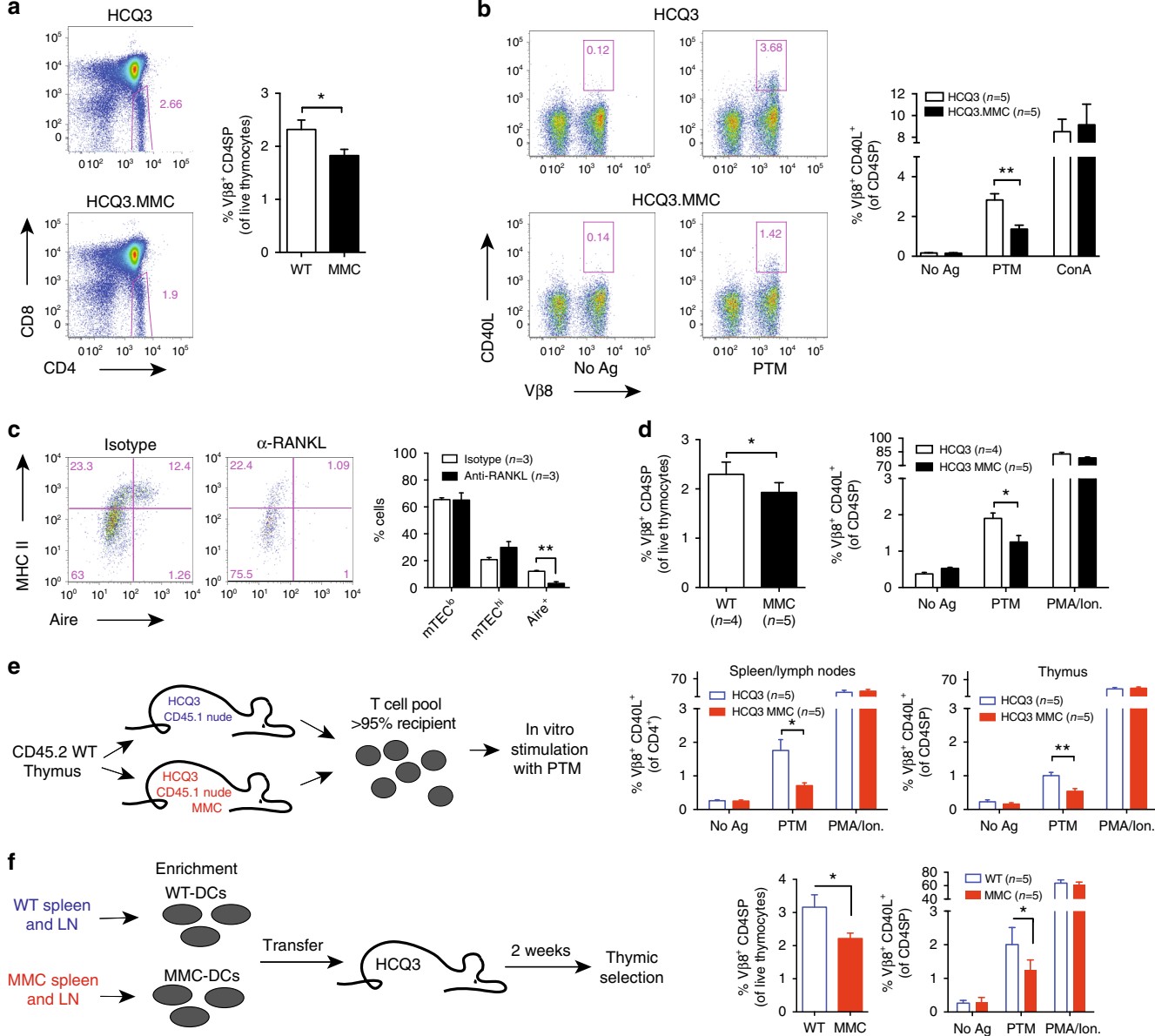

**Fig. 5** Peripheral antigen migration mediates thymic tolerance to the PTM epitope. Frequency of (**a**) CD4 single-positive and (**b**) CD4$^+$CD40L$^+$ thymocytes from naive HCQ3 and HCQ3.MMC mice, after antigen stimulation in vitro. Cells left unstimulated (No Ag) or concanavalin A (ConA) stimulated were used as negative and positive controls, respectively. **c** Treatment with anti-RANKL antibody induces selective depletion of Aire-expressing mTECs in the thymus. Indicated number of mice was treated with 100 μg of anti-RANKL or an isotype control antibody every second day for 2 weeks, and thymi were collected and analyzed the following week. Dot plots show representative examples of outcome following antibody treatment. **d** Treatment with anti-RANKL antibody does not alter central tolerance of HCQ3 transgenic T cells in HCQ3.MMC mice as determined by frequencies of CD4SP thymocytes (left) or up-regulation of CD40L following ex vivo stimulation with the PTM CII$_{260-270}$ peptide. Cells cultured in the absence of antigen (No Ag) and in the presence of PMA/ionomycin (PMA/Ion) were used as negative and positive controls, respectively. **e** Three- to four-week-old HCQ3-nude and HCQ3.MMC-nude mice were grafted with neonate thymus from wild-type mice. Thymus and pooled spleen and lymph nodes from individual mice were recovered 14–21 weeks after transplantation, when recipient-derived T cells constituted >95% of the peripheral T cell pool (as determined by CD45.1 expression) and investigated for up-regulation of CD40L, as described in **d**. **f** CD11c$^+$ cells were enriched from spleens and lymph nodes of naive MMC or WT donors and transferred to the indicated number of naive HCQ3 mice. Two weeks later, recipient mice were sacrificed and thymocytes were prepared and investigated ex vivo for frequency of CD4SP cells (left) and up-regulation of CD40L, as described for in **d**. Data from two pooled experiments are shown. $p$ values were calculated by unpaired $t$ test. *$p < 0.05$; **$p < 0.01$. Gating strategies used for analysis of flow cytometry data are shown in Supplementary Fig. 8

on Aire$^+$ mTEC. However, central tolerance to the PTM variant appears to be less efficient and does not occur via an Aire-dependent mechanism.

Migratory APC, particularly dendritic cells (DCs), contributes to central tolerance by acquiring and transporting peripheral and blood-borne antigens to the thymus[32,33]. To investigate whether the reduction of CD4SP thymocytes in HCQ3.MMC mice instead

depended on migratory APC, we established HCQ3.MMC-expressing mice on the nude background and transplanted these with a WT thymus. In this scenario, the MMC-derived CII$_{260-270}$ epitope will only be expressed in the periphery and cannot be expressed as an intrinsic TRA by the WT donor thymus. The grafted thymus allowed for egress of HCQ3-expressing thymocytes to the periphery where they displayed a tolerogenic

phenotype (Fig. 5e). Furthermore, thymocytes recovered from the grafted thymus of HCQ3.MMC-nude mice also displayed reduced levels of PTM-specific cells. These observations suggested that the PTM variant was transported from the periphery to the thymus, where it induced central tolerance.

Adoptive transfer of HCQ3 T cells to MMC recipients showed that the joint-derived PTM antigen is available for immune recognition in secondary lymphoid organs (Supplementary Fig. 6b). To investigate whether migratory DCs would be responsible for transport of the PTM epitope to the thymus, CD11c[+] DCs were enriched from spleens and lymph nodes of naive WT or MMC mice and transferred to HCQ3 recipients. Supporting our experimental approach was also the fact that subcutaneously injected fluorescent beads could be found in CD11c[+] cells in the thymus, days after injection (Supplementary Fig. 7). HCQ3 mice receiving DC from MMC donors displayed reduced levels of CD4SP thymocytes two weeks after cell transfer, in comparison to recipients of WT DC (Fig. 5f). Taken together, our data indicate that both the non-modified and PTM variants of the same self-antigen are presented in the thymus for tolerance induction. However, tolerance induction to these two variants is differently mediated: while the presentation of the native antigen is Aire-regulated, presentation of the PTM variant seems to depend on its transport to the thymus by peripheral APC.

## Discussion

The relevance of PTM antigens has been extensively discussed as a potential cause or perpetuation of autoimmunity (reviewed in ref. [34]). In fact, T cell reactivity to PTM antigens are frequently found in patients suffering from RA or type 1 diabetes[7,8]. However, unambiguous experimental validation of this scenario has been missing. Here, we demonstrate for the first time that central tolerance to non-modified and PTM variants of the same self-antigen is differently regulated. It is important to stress that the current study only addresses immunological tolerance to CII. Whether T cells specific for other PTM self-antigens escape central tolerance in a similar manner as described here needs to be addressed in subsequent studies.

Previous studies using TCR transgenic mice have made clear that thymic central selection plays an important role in achieving T cell tolerance[35–38]. Promiscuous presentation of TRA on mTEC through Aire, Fezf2, or yet unknown mechanisms, induces self-tolerance via negative selection and/or Treg induction in the nascent T cell repertoire[39–42]. It has also been demonstrated that peripherally expressed antigens can be transported to the thymus by migratory APC, and thus contribute to central T cell tolerance induction[32,33]. Despite the different roads leading to central tolerance, it has also been demonstrated that peripheral mechanisms can achieve non-responsiveness to self-antigens that are not available for presentation in the thymus[43,44]. The current study underscores the central role of the thymus in T cell tolerance induction, and it highlights two additional findings. First, we show that central tolerance mediated by mTEC is very robust, but limited to the non-modified version of a self-antigen. Second, negative selection of T cells specific to the PTM variant of the same antigen can only be achieved by transport of the peripheral antigen to the thymus by professional APC. Noteworthy, the degree of T cell tolerance to the PTM self-antigen version is apparently weaker in our model.

PTMs occur in a wide variety of proteins and are considered a mechanism of protein diversification. They are often generated through enzymatic activity and may alter solubility, activation, degradation or protein–protein interactions. Defects in PTM have been associated with developmental disorders and human diseases, underscoring the relevance of PTM in maintaining cellular

homeostasis[45]. However, from an immunological point of view, PTM constitute a challenge to the immune system, since they greatly increase the number of self-antigens against which tolerance must be established. Because some PTM only occur in specific cell lineages in a given tissue, we raised the hypothesis that some of these PTM may not be generated and/or presented in the thymus at a sufficient level as to establish T cell tolerance. Using CII as a model autoantigen that displays a critical involvement of PTM in T cell reactivity, we report here that tolerance towards the non-modified and PTM variants of the same antigen are independently and differentially regulated. We show that, in contrast to the non-modified antigen, the PTM variant obviously failed to be generated by mTEC in the thymus. However, the perceptible impact of the PTM epitope on intrathymic T cell selection could be traced back to its exclusive expression in peripheral tissues. Hence, peripheral expression of the PTM antigen is potentially able to establish tolerance via anergy or ignorance induction[46], as well as by negative selection and/or Treg induction of developing thymocytes. However, despite its availability in both peripheral and central lymphoid organs, PTM-specific autoreactive T cells persisted within the peripheral repertoire and allowed for development of autoimmunity (Fig. 2).

Theoretically, most self-antigens could be carried into the thymus by APC, and establish tolerance to newly generated/modified proteins. However, as shown here and earlier[47,48], as well as evidenced by the strong phenotype of autoimmune polyendocrine syndrome 1 patients, peripheral tolerance mechanisms and/or transport of peripheral self-antigens to the thymus cannot compensate for the lack of autochtonous thymic-induced central tolerance. A critical event in the development of MHC class II-associated autoimmune diseases is likely to be associated with T cell recognition of poorly tolerized PTM neo-epitopes. One example is thyroglobulin, which constitutes a target in autoimmune thyroiditis, and in which case PTM through iodination have been proposed as a mechanism leading T cells to escape tolerance[48]. Although thyroglobulin is available in the thymus for central tolerance induction, iodination does not occur in the thymus, and sufficient presentation of iodinated thyroglobulin in the thymus is unlikely to occur due to its very low blood levels. Consequently, T cells specific for thyroglobulin-derived peptides harboring iodinated tyrosine are likely to escape central tolerance, while iodinated thyroglobulin constitutes the physiological form in the tissue. In case of multiple sclerosis (MS), the autoreactive responses are though to be in part directed against myelin basic protein (MBP). The N-terminal part of MBP ($MBP_{1-11}$), where the alanine at position 1 is acetylated ($MBP_{Ac1-11}$), constitutes the primary target for encephalitogenic T cells in an animal model of MS[49]. In contrast, T cells do not respond to the non-acetylated $MBP_{1-11}$ peptide. Thymic expression of MBP protein is limited, but the $MBP_{1-11}$ sequence is readily available through the more widely expressed golli-MBP protein[50]. However, as golli-MBP expression is regulated by distinct upstream promoters, the $MBP_{1-11}$ sequence is not situated N-terminally in golli-MBP and therefore it is not presented in its acetylated form as to induce tolerance. Finally, with regard to the autoantigen studied here, the regulation of glycosylation is very complex and differs between different proteins, cell types, and species[51]. It is not clear how glycosylation of CII occurs, and how it is regulated enzymatically in vivo. CII is normally produced by chondrocytes, which have the capacity to hydroxylate and glycosylate lysine residues. However, analyses of recombinant CII protein, or $CII_{260-270}$ peptides produced from different cell types, have suggested that fibroblast-like cells, but not B cells, DCs or macrophages can produce galactosylated CII[52,53]. This clearly shows that the capacity for PTM of lysine residues is not a ubiquitous phenomenon. It is well documented that immune

 

responses to CII, including PTM with citrullination, oxidation and glycosylation, are frequently observed in RA patients[18,54–56]. Furthermore, T cell autoimmunity to CII is biased towards the glycosylated CII$_{260–270}$ PTM epitope in RA[18,56], implying that T cell tolerance to non-modified CII is strictly imposed. In this regard, the increased arthritis susceptibility observed in Aire-deficient MMC mice may at a first glance seem contradictory, as the non-modified form of the CII epitope is not a target structure in peripheral tissue. However, the rat CII used for immunization is derived from a Swarm rat chondrosarcoma that, in contrast to healthy joint-derived cartilage, also expresses the non-modified epitope (Fig. 1d). Following immunization, the more frequent T cells specific for the non-modified CII epitope in Aire-deficient mice may become activated and deliver help to B cell-producing CII-reactive antibodies that cross-react to mouse CII and thus induce clinical arthritis[57].

Our finding that T cell tolerance to distinct variants of the same self-antigen operates at different levels exemplifies a mechanism whereby autoreactive T cells can escape central tolerance induction. Although the generalization to other autoantigens needs to be further investigated, the scenario described for CII can in principal be applicable to any autoantigen which may be post-translationally modified, but where the capacity to perform these modifications is restricted to certain cell lineages, tissues, or environments. Importantly, PTM may also alter proteolysis of the self-antigens and consequent peptide sequence presentation and signal strength required for T cell selection and/or induction of thymic Tregs.

## Methods

**Mice.** All mouse strains used were on the C57BL/10.Q (B10.Q, or simply BQ) genetic background[58]. This strain expresses the MHC class II allele A$^q$, which presents the CII$_{260–270}$ peptide. Other genes have been introduced by backcrossing mice to B10.Q, and the respective littermates were used for experiments. The BQ.Ncf1 mouse strain differs from B10.Q simply by the Ncf1$^{m1j}$ mutation, which results in a reduced function of the NOX2 complex. Ncf1-mutated mice are thus highly susceptible to arthritis development[17]. Aire-deficient mice were obtained from The Jackson Laboratory on a C57BL/6 background, and were subsequently backcrossed onto B10.Q for six generations. The MMC transgenic mouse carries a mutated mouse CII gene in which the amino acid at position 266 had been altered from an aspartic acid to a glutamic acid, thereby expressing the rat/human CII$_{260–270}$ epitope in a cartilage-restricted manner[13]. MMC mice have been extensively backcrossed to B10.Q, and expressed the Ncf1 mutation since this allows the development of arthritis in MMC mice on the B10.Q background[17]. For the HCQ3 transgenic mouse, the TCR α- and β-chains were isolated from the previously described CII-specific HCQ3 T cell hybridoma clone[15], which displays high specificity for the galactosylated immunodominant T cell epitope of CII. The productively rearranged TCR Vα16-Jα10 and Vβ8-Jβ2.4 fragments were amplified by PCR from genomic DNA of HCQ3 cells, and inserted into the cassette vectors for expressing TCRα and TCRβ chains, respectively[59]. Finally, the vectors were linearized and co-injected into the pro-nucleus of B10.Q ES cells to generate the HCQ3 transgenic mouse on the B10.Q background. Screening for the α-cassette and the β-cassette allowed for the identification of a single offspring expressing both the transgenic α-chain and β-chain. This mouse was subsequently used as a founder for the HCQ3 transgenic strain. Athymic nude mice (Nu/J, H-2$^q$) and Rag1-deficient mice (B6.129S7Rag1<tm1Mom>/J) were obtained from The Jackson Laboratory. Rag1-deficient mice were crossed with B10.Q mice and litters further intercrossed to obtain mice with homozygote expression of A$^q$. All mice were bred and housed at the animal facility of Medical Inflammation Research. All experiments were done using littermates and followed the ARRIVE criteria[60]. All animals used were fed a standard rodent chow and given water ad libitum. Different experimental groups were housed together in order to minimize experimental bias. The local ethics committee approved all animal experiments (Stockholms Norra Djurförsöksetiska Nämnd, Stockholm, Sweden). All in vivo arthritis experiments as well as in vitro experiments using samples from laboratory mice were covered by the ethical numbers M107/07, N66/10, and N490/12. Anesthesia of animals was accomplished by isoflurane inhalation, whereas sacrifice was performed with CO$_2$.

**Antigens.** The rat CII was obtained from pepsin-digested SWARM, and subsequently processed as previously described[61]. CII peptides, containing the 259–273 sequence of rat CII with a non-modified lysine at position 264 or with a

[β]$_D$-galactopyranosyl residue on L-hydroxylysine at position 264 (PTM), were synthesized as previously described[62].

**Collagen-induced arthritis.** Mice were injected at the base of the tail with 100 μl emulsion consisting of 100 μg rat CII emulsified 1:1 in CFA (Difco, Detroit, MI, USA). Development of clinical arthritis was followed three times weekly through visual scoring of the paws, starting 2 weeks after immunization. The arthritis was scored using a scale ranging from 1 to 15 for each paw, with a maximum score of 60 per mouse. Each arthritic toe and knuckle was scored as 1, with a maximum of 10 per paw. A score of 5 was given to an arthritic ankle.

**Thymus transplantation.** Noninvasive thymus transplantation was performed as described by Basso et al[63]. Thymic lobes removed from newborn pups were immediately grafted into the axillary cavities of 3–4-week-old nude mice. The incision was closed with sutures, and the mouse was placed in a warm environment until it recovered from anesthetics. All mice were grafted with one entire thymus in one of the axillary cavities. Peripheral blood from transplanted mice was collected at regular intervals and samples were analyzed by flow cytometry. The establishment of a peripheral T cell pool was followed by monitoring the expression of CD45.1 (recipient-derived) and CD45.2 (donor-derived) on CD4 T cells obtained from peripheral blood (Supplementary Fig. 5). Mice were immunized with CII in CFA 11 weeks after transplantation, when more than 95% of the peripheral T cell pool was of recipient origin.

**Immunoassays.** ELISPOT assays were performed as previously described[64]. Briefly, mice were immunized with CII in CFA, and 10 or 70 days later cells were prepared from the spleen and draining lymph nodes and re-stimulated with CII peptides. For detection of interferon-γ (IFN-γ) spots, plates were pre-coated according to the manufacturer's instructions with R46A2 clone (10 μg ml$^{-1}$, from our in-house collection) and detection was achieved with biotinylated AN18.17.24 (Mabtech, Nacka Strand, Sweden). For IL-17A detection, TC11-18H10 (4 μg ml$^{-1}$; BD Biosciences, San Diego, CA, USA) and biotinylated TC11-8H4.1 (2 μg ml$^{-1}$; BD Biosciences) were used as capture and detection antibodies, respectively. Binding of biotinylated antibody was revealed by using streptavidin-conjugated alkaline phosphatase and the substrate Sigma Fast BCIP/nitroblue tetrazolium (Sigma).

T cell hybridoma responses were determined as described previously[65]. For adoptive transfer experiments, spleen cells were prepared from naive HCQ3 transgenic mice on a Rag1-deficient background and T cells were enriched through density centrifugation using Lympholyte (Cedarlane, Burlington, ON, Canada). CD4$^+$ T cells were isolated by negative selection (Dynal Mouse CD4 Negative Isolation Kit, Invitrogen, Eugene, OR, USA), washed, and then labeled with the fluorescent dye CFSE (Invitrogen) according to the manufacturer's protocol. Cells were washed twice with phosphate-buffered saline (PBS) and immediately injected intravenously in recipient mice.

**Qualitative and quantitative RT-PCR.** Complementary DNA (cDNA) was obtained by extraction (PureLink RNA Mini Kit, Ambion, Life Technologies) and conversion (High Capacity cDNA Reverse Transcription Kit, Life Technologies) of total RNA derived from thymi of Aire$^{Suf}$ and Aire$^{KO}$ mice. Gene-specific primers and probes were designed (Eurofins MWG Operon) for the mouse mRNA sequence (Ensembl Mouse, NCBI m37 assembly). Qualitative and quantitative RT-PCR was performed with optimized primers for detection of Col2a1 (forward primer 5′-CGACTGTCCCTCGGAAAAAC-3′; reverse primer 5′-GGAG-GAAAGTCATCTGGACGTT-3′; MGB probe 5′-TCCACTTCAGCTATGGC-3′), Ins2 (forward primer 5′-TTGTCAAGCAGCACCTTTGTG-3′; reverse primer 5′-AGCTCCAGTTGTGCCACTTGT-3′; MGB probe 5′-TGGAGGCTTCTTCTACCTG-3′), Gad67 (forward primer 5′-TCCAGTGCTCTGCCATTCTG-3′; reverse primer 5′-GCTTGTCTGGCTGGAAGAGGTA-3′; MGB probe 5′-AGGGTA-TACTCCAAGGAT-3′) and Ppia (forward primer 5′-CAGACGC-CACTGTCGCTTT-3′; reverse primer 5′-TGTCTTTGGAACTTTGTCTGCAA-3′; MGB probe 5′-CCCACCGTGTTCTT-3′), and 50 ng μl$^{-1}$ cDNA template were used in a 50 cycle reaction (7900HT Fast Real-Time PCR; Applied Biosystems, USA). A separate set of primers was used for qualitative RT-PCR to detect the CII$_{260–270}$ region of CII (forward primer 5′-GCGGGTGAACCTGGC-3′; reverse primer 5′-CTCGAGCACCTCGTTTGC-3′). Ins2 and Gad67 were used as controls for genes expressed either in the thymus alone or in both thymus and spleen, respectively (www.informatics.jax.org; ID MG:1205697 (Gad1/Gad67) and MG:1204189 (Ins2)). Gene expression Ct values were normalized to the housekeeping gene cyclophilin A (Ppia) and calibrated to a relevant sample for comparative analysis. Expression fold is presented as $2^{-\Delta\Delta Ct}$.

**Synthesis of a di-Lys-knot resin.** The synthesis of a di-Lys-knot resin: (Fmoc-Ahx)$_3$-Lys$_2$-Gly-Tyr-Lys(biotin)-Gly-NovaPEG Rink Amide resin was performed manually using a NovaPEG Rink amide resin (Novabiochem) with a loading of 0.18 mmol g$^{-1}$. All Fmoc deprotections were carried out with 20% piperidine in N-methyl-pyrrolidone (NMP) for 2 × 10 min. The coupling steps were performed in NMP with amino acid/HCTU (2-(6-chloro-1H-benzotriazole-1-yl)-1,1,3,3-tetra-methylammonium hexafluorophosphate)/DIPEA (N,N-diisopropylethylamine) (5:5:10) during 30 min to 1 h with occasional vortex. All Fmoc deprotections and

coupling steps were monitored using the Kaiser test[66] After incorporation of the second Gly residue and Fmoc deprotection, the orthogonally protected amino acids Fmoc-Lys(Mmt)-OH and Fmoc-Lys(Fmoc)-OH were coupled sequentially to the peptide resin. Then, the Mmt group was removed with dichloromethane/trifluoroacetic acid/triisopropylsilane (97:1.5:1.5) for 2 min. The solvent was removed by applying a flow of $N_2$ (g) and the step was repeated 7–10 times. The two Fmoc groups on the other Lys residue were deprotected with 20% piperidine in NMP and Fmoc-6-aminohexanoic acid (Ahx) was then coupled using amino acid:HCTU: DIPEA (15:15:20) to give the (Fmoc-Ahx)$_3$-Lys$_2$-Gly-Tyr-Lys(biotin)-Gly-Nova-PEG Rink Amide resin. This was then used directly in the synthesis of the naive and PTM triple helical CII peptides.

**Synthesis of native and PTM triple helical CII peptides.** The two triple helical peptides (THPs) were synthesized on a 20 μmol scale using the Fmoc strategy on a Prelude peptide synthesizer (Protein Technologies) using the pre-synthesized (Fmoc-Ahx)$_3$-Lys$_2$-Gly-Tyr-Lys(biotin)-Gly-NovaPEG Rink Amide resin as solid support. The three identical α-chains were assembled simultaneously using amino acid (5 equiv), HCTU (4.5 equiv), and DIPEA (10 equiv) with respect to each amine, and coupled for 2 × 20 min. For THPGal, the automated synthesis was paused at position 264. (5 R)-$N^\alpha$-(Fluoren-9-ylmethoxycarbonyl)-$N^\varepsilon$-benzylox-ycarbonyl-5-$O$-(2,3,4,6-tetra-$O$-acetyl-β-D-galactopyranosyl)-5-hydroxy-L-lysine[67,68] (1.5 equiv) was activated with HATU (2-(7-Aza-1$H$-benzotriazole-1-yl)-1,1,3,3-tetramethyluronium hexafluorophosphate) (1.4 equiv) and DIPEA (4 equiv) and coupled manually for 3 h. Following Kaiser test to verify complete coupling, the THP synthesis was continued using the peptide synthesizer. Fmoc deprotection after each cycle was accomplished by treatment with 20% piperidine in NMP for 2 × 5 min. The glycopeptide THPGal was cleaved from the resin with a mixture of trifluoroacetic acid/thioanisole/$H_2O$/triisopropylsilane/ethanedithiol (94:2:2:1:1 (v v$^{-1}$)) for 3 h at 40 °C, using 50 ml mixture per gram of resin. THPK was cleaved with trifluoroacetic acid/$H_2O$/triisopropylsilane (95:2.5:2.5) for 2 h at ambient temperature. After filtration and evaporation of TFA by $N_2$ (g) bubbling, the peptide was precipitated and washed three times with cold diethyl ether (Et$_2$O), followed by lyophilization.

The crude peptide was dissolved in 0.1 M acetic acid (4 ml per 20 μM batch) and purified with reverse-phase high-performance liquid chromatography (HPLC) (Varian 940-LC) using a semi-preparative Grace Vydac C8-column (22 mm × 150 mm, 10 μm, 300 Å) with a gradient of ACN (acetonitrile):$H_2O$, containing 0.1% trifluoroacetic acid, from 10% (2 min) to 40% ACN in 20 min at a flow rate of 15 ml min$^{-1}$. The fractions containing the THP (verified by matrix assisted laser desorption/ionization-time-of-flight mass spectrometry (MALDI-TOF-MS)) were pooled, concentrated, and lyophilized. THPGal was deacetylated with NaOMe in MeOH (20 mM, 1 ml mg$^{-1}$ peptide) for 2–3 h at room temperature (monitored by analytical reverse-phase HPLC). Neutralization was achieved by the addition of acetic acid (9–10 ml) and the solution was then concentrated under reduced pressure, after which the residue was purified using reverse-phase HPLC followed by lyophilization. The yields of the final THPs were 7.7% and 15.0%, respectively, for THPGal and THPK.

The identity of THPGal and THPK was verified by MALDI-TOF-MS (Voyager-DE Pro, Applied Biosystems) using a sinapic acid matrix and with detection in the positive mode. MS $m/z$ calcd for THPGal [M+H]$^+$ 16,626.61, found 16,624.86. MS $m/z$ calcd for THPK [M+H]$^+$ 16,091.96, found 16,094.11. Circular dichroism (CD) spectroscopy was used, both to verify the triple helical structure of the two THPs and to analyze their thermal stability, i.e., their melting temperature ($T_m$). Both THPGal and THPK showed triple helical conformations at 20 °C, characterized by a maximum wavelength around 222 nm and a minimum wavelength around 195 nm. The thermal transition curves were obtained by recording the ellipticity (millidegrees) at 222 nm while the temperature was continuously increased over a range of 5–85 °C at a rate of 1 °C per minute. The inflection point in the transition region in the obtained sigmoidal melting curves was defined as the melting temperature ($T_m$). $T_m$ (THPGal) 52.5 °C. $T_m$ (THPK) 51.7 °C. The chemical structure of the triple helical CII peptides is presented in Supplementary Figure 5a.

**Generation of a mAb specific for PTM CII.** A B10.Q mouse was immunized twice (7 weeks apart) with 50 μl of an emulsion containing 20 μg of the triple helical CII$_{259–273}$ peptide harboring a galactosylated hydroxylysine at position 264 (THPGal) in incomplete Freund's adjuvant (IFA). Inguinal lymph node cells were prepared 5 days after the second immunization and fused with myeloma cells (P3X63-Ag8.653) as previously described[69]. Supernatants from growing cultures were tested for binding to ELISA (enzyme-linked immunosorbent assay) plates coated with rat CII (10 μg ml$^{-1}$), THPGal (2.5 μg ml$^{-1}$), and the control triple helical CII peptide harboring a non-modified lysine at position 264 (THPK; 2.5 μg ml$^{-1}$). Cells in wells positive for binding to both CII and THPGal but negative for THPK were expanded and further sub-cloned. After four additional rounds of screening and sub-cloning, one clone specific for CII and the THPGal peptide of the IgG2b isotype was obtained, and denoted T8.

**Histology and cytospin.** Thymi from 1-week-old to 2-week-old mice were harvested without blood contamination and cut into small pieces in RPMI. Thymocytes were released from the tissue by gently stirring the organ with a magnet rod

for 10 min. Thymi were then digested using collagenase IV/dispase/DNase in RPMI (200/200/25 μg ml$^{-1}$) in rounds of 15 min at 37 °C, until full digestion was achieved. An equal volume of 10 mM EDTA solution was added to the resulting supernatant after each digestion round. The resulting cell suspension was then enriched for CD45-negative cells by magnetic beads (Miltenyi Biotech). The obtained cells were diluted accordingly and spun onto microscope slides (Superfrost, VWR). Staining of cells was performed in PBS containing 1% bovine serum albumin and 0.5% Triton X-100 with UEA-1 (FITC labeled, Vector Laboratories, Burlingame, CA, USA), anti-CII (in-house produced mAb CIIC1 and CIIC2 biotinylated), and anti-MHC class II (clone M5/114.15.2, Alexa Fluor 647 conjugated, BioLegend) antibodies overnight at 4 °C. Slides were washed and incubated with streptavidin-Alexa568 (Invitrogen, USA), followed by a new washing step and mounted in DAPI (4',6-diamidino-2-phenylindole) containing media. Pictures were acquired using a Zeiss LSM700 (Carl Zeiss AG) confocal microscope.

Human thymus was obtained from three children undergoing corrective cardiac surgery. Parents gave informed consent, and the study was approved by the Regional Ethical Board at the University of Gothenburg, Sweden (no. 217-12, 2012-04-26). Human thymus tissue was imbedded in OCT and cut to 7 μm sections with a cryostat. The sections were fixed with cold acetone and blocked with Protein block (X0909, Dako) with 5% donkey serum (D9663, Sigma). The tissue was stained in 4 °C for 1 h with a cocktail of four anti-CII mAbs (clones CllC1, CllC2, UL-1, and M2139, in-house produced and biotinylated) and anti-AIRE (sc-17986, Santa Cruz Biotechnology). The tissues were washed and incubated in 4 °C for 30 min with Streptavidin Alexa Fluor 635 (S32364, Life Technologies), anti-goat donkey Alexa Fluor 555 (A21432, Life Technologies), and Hoechst (H21486, Life Technologies). Sections were mounted with Vectashield (H-1000, Vector Laboratories) and images acquired using a LSM700 (Carl Zeiss AG).

Whole thymi, nose, and paws from 4-day-old WT and MMC mice were collected and kept in OCT. Tissues were cut in 7 μm sections and fixed with cold acetone. Anti-CII cocktail antibody stain and negative control stain (M2139-S31R; a single point mutation at amino acid 31 results in abrogation of CII recognition by the M2139 mAb[70]) were performed as described above. For the PTM CII stain, slides were rehydrated in PBS, treated with hyaluronidase (H3506, Sigma-Aldrich) for 20 min at 20 °C and 6 M urea, and 0.05 M sodium acetate (pH 4.8) at 50 °C for 20 min. Tissue sections were blocked with protein block (X0909, Dako) and stained with a PTM CII-specific mAb (clone T8) for 1 h, followed by streptavidin Alexa Fluor 555 (S21381, Life Technologies) for 30 min. The slides were mounted with ProLong gold (P36930, Life Technologies).

For the isolation of human mTECs, human thymus tissue was digested mechanically and enzymatically (DNase I, 11284932001, and Liberase TH, 5401127001, Roche) to a single-cell suspension, density centrifuged (Percoll 1.07 g ml$^{-1}$, 17-0891-01, GE Healthcare Life Sciences) and stained with anti-CD45 (560178, BD Bioscience), anti-EpCAM (324206, BioLegend), anti-HLA-DR (347403, BD Bioscience), and CDR2 (HB-214, ATCC), and FACS sorted with an SY3200 (Sony). Cytospins of human samples were prepared from the sorted mTECs defined as CD45$^-$EpCAM$^+$CDR2$^-$HLA-DR$^{lo/hi}$ for immunohistochemistry. The samples were stained and mounted as described above.

**Anti-RANKL treatment.** Depletion of Aire$^+$ mTECs was done as previously described[31]. Briefly, mice were injected intraperitoneally every other day for 2 weeks with 100 μg of anti-RANKL mAb (clone IK22/5) or isotype control (clone 2A3; BioXcell, West Lebanon, NH, USA). After 2 weeks, mice were sacrificed and thymi harvested. Organs were enzymatically digested as described above for mouse thymus. TECs were enriched by negative anti-CD45 labeling (Miltenyi Biotec, Boston, MA, USA) and further stained for flow cytometry analysis.

**Flow cytometry.** The following anti-mouse antibodies and staining reagents were used: anti-Vβ8 FITC (clone F23.1), anti-CD40L PE (clone MR1), anti-CD80 PE (clone 16-10A1), anti-CD45 PE/Cy7 (clone 30-F11), anti-CD8 Brilliant Violet 650 (clone 53-6.7), and biotinylated anti-Ly51 from BD; anti-CD4 Pacific Blue (clone RM4-5), anti-Ep-CAM APC (clone G8.8), and anti-IA/IE Pacific Blue (clone M5/114.15.2) from BioLegend; anti-Aire FITC (clone 5H12) from eBioscience; Qdot 655-Streptavidin conjugate and LIVE/DEAD® Fixable Near-IR Dead Cell Stain from Life technology as well as anti-FcγRIII/II (clone 2.4G2) in-house produced. For detection of CD40L, single-cell suspensions were processed as described earlier[65]. In brief, $1.2 \times 10^7$ cells ml$^{-1}$ were cultured for 6 h in the presence of the peptides (10 μg ml$^{-1}$), anti-CD28 (1 μg ml$^{-1}$, clone 61109; R&D Systems, Minneapolis, MN, USA), and brefeldin A (2 μg ml$^{-1}$, Sigma-Aldrich, St. Louis, MO, USA). Cells were then washed, Fc blocked with unlabeled 2.4G2 antibody, and stained at room temperature with surface markers. Cells were then fixed and permeabilized with Cytofix/Cytoperm solution (BD Pharmingen, San Diego, CA, USA) and stained for CD40L, according to the manufacturer's instructions. Flow cytometry acquisition was performed in an LSR-II (BD Biosciences) and analyzed using FlowJo software (TreeStar Inc., Ashland, OR, USA). Gating strategies used for analysis of flow cytometry data are shown in Supplementary Fig. 8.

**Statistical analysis.** Statistical calculations were done with a statistical software package (GraphPad Prism 5.0b). Antibody levels, in vitro lymphocyte assays,

arthritis severity, and quantitative PCR data were analyzed with Mann–Whitney U test. Frequencies of populations determined by flow cytometry were analyzed using Student's t test. Arthritis incidence was analyzed with Fisher's exact test.

**Data availability**. The data that support the findings of this study are available from the corresponding author upon request.

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

## Acknowledgements

This study was supported by The Swedish Strategic Science Foundation, Knut and Alice Wallenberg foundation, Swedish Research Council, and the EU Innovative Medicine initiative BeTheCure grant. B.R. was supported by Konung Gustaf V:s 80-årsfond. We thank Carlos and Kristina Palestro for taking excellent care of the experimental animals.

## Author contributions

B.R. designed experiments, conducted animal studies, cellular assays, data analyses and interpretations, and wrote the manuscript. P.M. performed animal studies, cellular assays, data analyses, and interpretations. H.Y. generated the TCR transgenic mouse and assisted in data interpretations. V.U. performed cellular assays, data analyses, and interpretations. C.L. performed immunofluorescence on sections and sorted cells from human thymus. C.N. performed confocal microscopy of murine samples and helped in data analyses. J.V. synthesized and characterized the THPs. J.K. synthesized and characterized the THPs. B.K. contributed with manuscript preparation and data interpretation. O.E. contributed with human samples, data analyses, and interpretations. R.H. initiated and designed the project, was co-supervisor, and helped writing the manuscript. J.B. designed the project, conducted animal experiments and cell assays, compiled and analyzed data, co-supervised the project. and wrote the manuscript.

## Additional information

**Competing interests:** The authors declare no competing financial interests.

