## [Peer Review File · Nature Communications]

Reviewers' comments:

Reviewer #1 (Remarks to the Author):

The manuscript by Raposo, et al., addresses several key questions regarding immune tolerance to posttranslationally modified self proteins. Overall, this topic is timely and the work is novel and well-supported by the data provided. Regarding the topic, PTM self proteins continue to be studied in the context of many specific autoimmune syndromes and provide insights into both novel immune targets as well as serving as biomarkers of the onset, progression, and/or severity of disease. The unknown features about these pathways include questions that are experimentally pursued in this manuscript, notably, what are the potential mechanisms to establishing tolerance (or lack of tolerance) to PTM self. The models are generally robust, of clever design (MMC strains, thymic transplant and DC trafficking studies, T cell specificity, etc) and supportive of the discussion and conclusions provided in the manuscript. The central conclusions are important although they may not explain tolerance induction to all types of PTM self proteins. In particular, the CII glycosylated determinant is dependent on an enzyme mediated process. These levels can be carefully and clearly defined in experimental models (amounts required to elicit a response, or pathology). In contrast, spontaneous modifications such as oxidation, carbonization, or isoaspartyl modifications that arise without direct enzyme interactions with substrate may be more difficult to fit in to this overall model. For example, the latter spontaneous PTMs are often driven in specific compartments as a response to inflammation. Their levels, and thus amounts that alter tolerance, are likely very different from this model system. PTM self are known for being alternatively processed by the immune system. That is, PTMs may alter how APCs process and present particular determinants (CII in this system). This reviewer would appreciate discussion regarding processing (and/or MHC binding changes) that may be invoked by the PTM CII, as this also reflects on the ability of various iso forms of 'self' to cause thymic or peripheral tolerance or activate T cells in various figures. Moreover, several figures stimulate T cells with PTM or native CII peptide. Is it impossible for native protein (or peptide) to become glycosylated in vivo and trigger T cell responses? Specific experiments are not necessary, as various controls provided in the paper do illustrate that both forms can be presented by MHC. Finally, the reference list needs to be checked carefully. Ref 48 (line 312 appears incorrect), ref 49 (line 322), ref 50 (line 324). In summary, the work is carefully performed with appropriate controls. The data support conclusions regarding tolerance to PTM self proteins. The construction of the model allows for sensitive and accurate interpretations, something that has evaded other disease associated PTM auto antigens. For example, the parallel processes in rodent and human CII responses cannot be (and have not been) addressed in a similar manner with other human PTM autoantigens (such as those found in T1D, MS, or SLE). In this regard, the work provides clear insights and advances in understanding tolerance to peripheral PTM self antigens.

Reviewer #2 (Remarks to the Author):

Referee's comments to Raposo et al. 2017

CP 20170516

The study by Raposo et al., argues that T cells specific to post-translationally modified (PTM) self-antigens escape thymically-induced tolerance and subsequently promote autoimmunity. The question whether tolerance to PTM proteins is induced centrally in the thymus represents a major and still enigmatic question in the field.

In particular, the authors investigate the tolerance induction to type II collagen (CII) in an autologous collagen-induced arthritis mouse model. The hallmark of arthritis is the production of autoantibodies directed against PTM (mainly citrullinated) proteins (e.g. collagen) and/or autoantibodies against rheumatoid factor. Autoantibodies are presented in the serum and joints in high titers.

Using the above mentioned mouse model, the authors demonstrate that tolerance to CII in the

thymus is restricted to the native (non posttranslational modified) variant of CII and it is Aire dependent. In contrast, tolerance to the PTM variant of CII is not mediated by thymus-intrinsic, Aire-dependent mechanism, but rather through import of this PTM CII variant to the thymus by peripheral antigen presenting cells.

Major issues:

While the concept of the paper is interesting and has further important implications, its major issue is that the authors derive very general conclusions based on a single antigen. Such generalization is very dangerous as other (and maybe even most self-antigens) might be tolerized in the thymus.

Therefore to fully support their statements and conclusions, the authors would have to validate additional examples of self antigens, preferably associated with other autoimmune disorders (e.g. type-1 diabetes associated antigens which are expressed in the thymus), which may prove quite challenging.

Moreover, due to the technical difficulties of assessing arthritis development in nude mice, all autoimmunity assays were done in-vitro, which could give rise to artifacts outside of the physiological context (While in fact the actual autoimmunity caused by these cells could be mild).

In addition, there is no clear quantification of the native and the PMT CII variants in the thymus and the joint of the MMC mice. Quantification of these individual variants would certainly help to better support the authors' conclusions.

Although the manuscript provides some mechanistic explanations about the tolerance induction to the PTM CII variant, the authors do not demonstrate if tolerance induction in their model is due to negative selection and/or Treg induction. This would be an important aspect to address.

Finally, the transfer of bulk DCs isolation from WT mice or MMC mice leads to the changes in HCQ3 SP CD4 T-cells frequency (Fig. 5f). However, in this particular setting, there is no direct evidence provided, that DCs are able to bring Col2a1 as the antigen directly into the thymus. The observed change in T-cell frequency can be viewed as the consequence of immune reaction in the periphery or perturbation in the immune system after the transfer. Probably, the intrathymic injection of the WT and PTA peptide should be tried. Alternatively, for instance CFSE-labeled DCs can be pulsed with WT or PTA peptide and transfer to HCQ3 mice. Their migration of DCs into the thymus can be monitored, as well as their impact on HCQ3 T-cells selection.

Minor issues:

1. In general, I would suggest using scatter plots instead of bar graphs in order to visualize better values distribution.
2. Since fibroblasts can also express different types of collagens, it would be beneficial to determine if mTECs are the only source of Col2a1 expression within the thymus by FACS-sorting of thymic fibroblasts, mTECslow, mTECshigh, and possibly other thymic populations and subsequent qPCR analysis.
3. In page 6 line 122 "differentlually" should be changed for "diferentially".
4. Term "data not shown" (Line 147 page 7 & line 226 page 10) should be excluded, instead data should be provided as supplementary information.

5. There is a lack of clarity in lines 155-160. It is not clear enough how this system would allow to know if negative selection of CII specific T cells results from the presentation of the non-modified CII or an unrelated auto-antigen. The authors might consider rewriting those sentences.
6. The statement Col2a1mRNA transcript was detected only in the thymus but not in other lymphoid organs is rather inaccurate (pg. 6-7, 145-147). As the corresponding figure shows only the absence of Col2a1mRNA transcript in the spleen. The statement should be rephrased or PCR detection of Col2a1 expression in other lymphoid organs should be included.
7. Mice carrying the Ncf1 mutation are well described in methods section but they are not even mentioned in the results section, however the Fig.1 is completely based on them. Please include the rationale of using this particular strain directly into the result section.
8. Fig.1 – Ncf vs. Ncf1. The strain name should be used consistently in graph legends.
9. In line 110-112 (referring to Fig 1b) authors state that in MMC mice response to the native form is almost abrogated in comparison with the PMT form. What was the statistical test applied? This information should be included. If what was tested is MMC mice response to native vs. PTM, Wilcoxon should be used instead of Mann Whitney.
10. To support that MMC mice are less susceptible to arthritis (Fig1c) it would be valuable to include the histology of the cartilage or pictures of arthritic limbs showing the severity of the lesions. (This also applies to Fig 2a)
11. In line 131 “mice mounted a significant response only...” was there a statistical evaluation performed? If not, the word “significant” is not justified.
12. In figure legend 3 a, the authors do not provide the information about the displayed data. Information regarding whether the bars show SD or SEM etc. should be provided.
13. Fig. 3 and 5 are rather blurry and their resolution is low.
14. Immunofluorescence images (Fig 3 and Supplementary figure 4) should be stained with Aire, CII and some cytoplasmic or cell surface marker in order to correlate Aire and CII protein presence in mTECc and to verify the localization of CII within mTECs.
15. In figure 5 a and b authors show reduction in the frequency of CD4 and CD4 CD40L+ cells in HCQ.MMC mice in comparison with HCQ mice. It would be valuable to add cell counts as well as to check if Treg compartment is affected
16. Instead of no antigen control the irrelevant peptide control (e.g. OVA) should be used in antigen proliferation or activation experiments using CII specific HCQ3 T cells (Fig 5).
17. The Suppl. Fig. 6 presents proliferation of HCQ3 cells in MMC mice after their transfer. Are HCQ3 T-cells itself able to induce arthritis in MMC mice? If not, is the transfer of HCQ3 cells to T-cell deficient animal model crossed with MMC mice able to promote it?

Reply to Reviewers

Reviewer #1 (Remarks to the Author):

The manuscript by Raposo, et al., addresses several key questions regarding immune tolerance to posttranslationally modified self proteins. Overall, this topic is timely and the work is novel and well-supported by the data provided. Regarding the topic, PTM self proteins continue to be studied in the context of many specific autoimmune syndromes and provide insights into both novel immune targets as well as serving as biomarkers of the onset, progression, and/or severity of disease. The unknown features about these pathways include questions that are experimentally pursued in this manuscript, notably, what are the potential mechanisms to establishing tolerance (or lack of tolerance) to PTM self. The models are generally robust, of clever design (MMC strains, thymic transplant and DC trafficking studies, T cell specificity, etc) and supportive of the discussion and conclusions provided in the manuscript.

The central conclusions are important although they may not explain tolerance induction to all types of PTM self proteins. In particular, the CII glycosylated determinant is dependent on an enzyme mediated process. These levels can be carefully and clearly defined in experimental models (amounts required to elicit a response, or pathology). In contrast, spontaneous modifications such as oxidation, carbonization, or isoaspartyl modifications that arise without direct enzyme interactions with substrate may be more difficult to fit in to this overall model. For example, the latter spontaneous PTMs are often driven in specific compartments as a response to inflammation. Their levels, and thus amounts that alter tolerance, are likely very different from this model system. PTM self are known for being alternatively processed by the immune system. That is, PTMs may alter how APCs process and present particular determinants (CII in this system).

RI-Q1: *This reviewer would appreciate discussion regarding processing (and/or MHC binding changes) that may be invoked by the PTM CII, as this also reflects on the ability of various isoforms of 'self' to cause thymic or peripheral tolerance or activate T cells in various figures.*

We agree that this is an interesting question, although a possible differential processing due to the particular PTM will not affect the conclusions of this manuscript. However, we have earlier investigated this issue in more detail, and taken together we found that the PTM (i.e. galactosylation, which is the major T cell epitope) does not affect antigen processing *per se* and does not affect binding of the peptide to the MHC-II molecule. This has been described in several publications using T cell hybridomas that originated from mice suffering from collagen-induced arthritis [CIA; (1-7)]. The immunodominant epitope of CII (CII₂₆₀₋₂₇₀) can be PTM by hydroxylation of K264, which in turn is glycosylated with a galactose moiety, followed by a glucose residue. All forms of CII occur in cartilage, *in vivo* (native, hydroxylated, mono- and disaccharide), although CII is predominantly glycosylated (7). The major MHC-II contact points are the I260 and F263 residues, whereas the K264 is not involved in MHC binding. Moreover, there are no detectable modifications of the PTM at K264 after antigen processing *in vivo* or *in vitro*. These observations, together with the experiments presented in the present manuscript,

suggest that the presence of the antigen, rather than its TCR signaling strength, dictate T cell selection to the PTM CII.

Nevertheless, this does not exclude the fact that, for other PTM self-antigens, PTMs may affect proteolysis of the protein and consequent peptide sequence presentation and signal strength required for T cell selection/activation.

Following the advice of the reviewer, we have now extended and clarified this point in the discussion.

RI-Q2: *Moreover, several figures stimulate T cells with PTM or native CII peptide. Is it impossible for native protein (or peptide) to become glycosylated in vivo and trigger T cell responses? Specific experiments are not necessary, as various controls provided in the paper do illustrate that both forms can be presented by MHC.*

CII is believed to be produced mainly in chondrocytes and it has been unclear to what extent other cells can produce and export CII. Glycosylation of CII in the chondrocyte takes place soon after translation and before the triple helix conformation is completed. The glycosylation process is dependent on complex and unique set of enzymes in order to occur. It has so far been believed that such set of enzymes only occur in chondrocytes. CII also occurs in the eye, where it is a component of the vitreous body, a compartment excluded from the immune system. Whether CII is glycosylated or not in the eye it is not known. In an artificial situation, fibroblasts transfected with a lentivirus encoding CII were shown to activate PTM CII₂₆₀₋₂₇₀ reactive hybridomas (HCQ3). On the other hand, DCs transfected in the same way failed to activate HCQ3 cells but instead were able to activate native CII₂₆₀₋₂₇₀ reactive hybridomas (8). These observations clearly show that the capacity to induce PTM on a given protein is restricted to certain types of cells. Along this line, glycosylated CII only occurs *in loco* at the cartilage, and at steady state the cartilage CII is predominantly glycosylated (7). The presence of PTM CII in lymph nodes of naïve mice, which most likely originate from draining of cartilaginous joints, has been demonstrated in this manuscript (supplementary figure 6). This suggests that at steady state, PTM CII is likely to constitute an antigen for peripheral tolerance. In fact, MMC mice are tolerized to CIA, and our data strongly indicates that such tolerance is mediated by peripheral mechanisms, as central tolerance plays no role in this disease model.

RI-Q3: *Finally, the reference list needs to be checked carefully. Ref 48 (line 312 appears incorrect), ref 49 (line 322), ref 50 (line 324).*

All references have now been checked and corrected. Due to the introduction of new references in the revised text, these numbers no longer apply.

In summary, the work is carefully performed with appropriate controls. The data support conclusions regarding tolerance to PTM self proteins. The construction of the model allows for sensitive and accurate interpretations, something that has evaded other disease associated PTM auto antigens. For example, the parallel processes in rodent and human CII responses cannot be (and have not been) addressed in a similar manner with other human PTM autoantigens (such as those found in T1D, MS, or SLE). In this regard, the work provides clear insights and advances in

understanding tolerance to peripheral PTM self antigens.

-/-

*Reviewer #2 (Remarks to the Author):
Referee's comments to Raposo et al. 2017
CP 20170516*

The study by Raposo et al., argues that T cells specific to post-translationally modified (PTM) self-antigens escape thymically-induced tolerance and subsequently promote autoimmunity. The question whether tolerance to PTM proteins is induced centrally in the thymus represents a major and still enigmatic question in the field.

In particular, the authors investigate the tolerance induction to type II collagen (CII) in an autologous collagen-induced arthritis mouse model. The hallmark of arthritis is the production of autoantibodies directed against PTM (mainly citrullinated) proteins (e.g. collagen) and/or autoantibodies against rheumatoid factor. Autoantibodies are presented in the serum and joints in high titers.

Using the above mentioned mouse model, the authors demonstrate that tolerance to CII in the thymus is restricted to the native (non posttranslational modified) variant of CII and it is Aire dependent. In contrast, tolerance to the PTM variant of CII is not mediated by thymus-intrinsic, Aire-dependent mechanism, but rather through import of this PTM CII variant to the thymus by peripheral antigen presenting cells.

Major issues:

R2-Q1: *While the concept of the paper is interesting and has further important implications, its major issue is that the authors derive very general conclusions based on a single antigen. Such generalization is very dangerous as other (and maybe even most self-antigens) might be tolerized in the thymus.*

Therefore to fully support their statements and conclusions, the authors would have to validate additional examples of self antigens, preferably associated with other autoimmune disorders (e.g. type-1 diabetes associated antigens which are expressed in the thymus), which may prove quite challenging.

As explained above, in order to extend our observations to another autoimmune disease it would be required to develop unique analysis and tools, which would indeed be quite challenging. Importantly, there are no other tools or possibilities to address this problem in other diseases or disease models, since this would require the precise knowledge of a relevant auto-antigen, isolation of T cells specific for the PTM and tools for detection of this PTM both in the thymus and target tissue. Thus, it is not presently possible to address this issue in other diseases. Nevertheless, we do not see a reason for why these proposed mechanisms should not be considered generally applicable.

We have carefully stated that our observations are true for CII and raise the hypothesis that self-reactive T cells to other PTM self-antigens may escape central tolerance via a similar mechanism. However, we extensively discuss in the

manuscript the relevance of other autoimmune PTM self-antigens (namely MBP_{Ac1-11} and iodinated thyroglobulin), and how the existing literature indicates that T cell reactivity to these PTM variants is associated with their absence in the thymus. Our results with PTM CII suggest that other PTM self-antigens may escape central tolerance in a similar way, such as MBP_{Ac1-11} and iodinated thyroglobulin. However, we do not claim it to be the solely answer to how PTM-dependent autoimmunity evolves.

R2-Q2: *Moreover, due to the technical difficulties of assessing arthritis development in nude mice, all autoimmunity assays were done in-vitro, which could give rise to artifacts outside of the physiological context (While in fact the actual autoimmunity caused by these cells could be mild).*

In figure 1 we show that MMC mice display T cell tolerance to self-CII and that this is associated with a strong, but incomplete protection from arthritis. In figure 2 we show that arthritis protection can be abrogated by Aire deficiency, and that this break of disease protection is accompanied by loss of T cell tolerance to the naïve T cell epitope. Together these experiments suggest that T cell tolerance to the naïve and glycosylated T cell epitope is differently regulated. We therefore decided to use Nude mice to elucidate where tolerance to the two different forms of the epitope is induced, by either grafting an MMC thymus into a normal Nude mouse, or by grafting MMC transgenic Nude mice with a wild type thymus. Hereby we can show that T cell tolerance to the naïve T cell epitope is established in the thymus, with little or no influence from the periphery, while T cell tolerance to the glycosylated epitope originates from the periphery. Hence the purpose was not to evaluate arthritis but to anatomically locate the origin of the T cell tolerance towards the two forms of the T cell epitope.

In figure 5, we used Nude mice expressing the HCQ3 TCR transgene, with or without co-expression of MMC, and grafted them with a wild type thymus. This experiment was designed in order to assess the frequency of CII-specific thymocytes in naïve animals. Hereby we can show a small but significant effect on thymic selection of these cells that must have originated from the periphery. These analyses were done by determining the frequency of CD40L expression following a few hours stimulation with antigen *ex vivo*.

Although it could be possible and interesting to perform arthritis experiments in grafted Nude mice in parallel to the experiments described above, it was not required in order to investigate the issues addressed. Furthermore, development of arthritis in A^g expressing mice is strongly dependent on B cells and it is quite uncertain to what extent the establishment of a peripheral T cell repertoire in grafted Nude would allow for physiological arthritogenic immune response to develop and that would resemble what we observe in MMC mice on a wild type background.

R2-Q3: *In addition, there is no clear quantification of the native and the PMT CII variants in the thymus and the joint of the MMC mice. Quantification of these individual variants would certainly help to better support the authors' conclusions.* We agree that this would strengthen the manuscript. We have therefore now used a new and previously unpublished monoclonal antibody specific for the galactosylated K264 at the CII₂₆₀₋₂₇₀ epitope, with which we have extended and quantified our

analysis in both mouse and human tissues. The data has been included in figure 3 as well as in supplementary information. Using both wild type and MMC littermates, we could clearly identify CII in the cartilage of the joint and nose, as well as in the thymus (mAb cocktail containing CIIC1, CIIC2, UL1 and M2139 clones). With the new in-house generated mAb specifically recognizing the galactosylated form of CII (mAb T8, described in *Methods* section), we could stain nasal and joint cartilages, confirming previous observations that cartilage CII is predominantly glycosylated (7). Importantly, all thymic samples were negative for T8 stain. As a negative control, we have stained the same tissues with a CII specific mAb (M2139) that has been single-point mutated [M2139-S31R; (9)] resulting in complete absence of CII recognition. None of the tissues was positively stained with M2139-S31R. Furthermore, we used keratin 5 to demonstrate co-localization of thymic CII expression within mTECs (figure 3). We have included the quantification of CII⁺ cells in the thymus of murine and human samples (supplementary table 1). This confirms that native CII, but not glycosylated CII, is expressed in mTECs and by a very limited fraction of mTECs.

R2-Q4: *Although the manuscript provides some mechanistic explanations about the tolerance induction to the PTM CII variant, the authors do not demonstrate if tolerance induction in their model is due to negative selection and/or Treg induction. This would be an important aspect to address.*

The induction of Treg cells in this model is a very interesting question, and we have considered the topic in the discussion of the manuscript. In fact, MMC mice have a strong tolerance to CIA, and we have shown in this manuscript that such tolerance occurs in the absence of thymic selection to PTM-reactive T cells. The mechanisms of peripheral tolerance involved in this disease model cannot be, therefore, disregarded. We have earlier published some observations that several different mechanisms (deletion, anergy and induction of regulatory T cells) are involved in CIA in both WT and MMC mice (3, 10-12). The precise mechanisms of tolerance are likely to be complex and therefore out of the scope of the present manuscript. In fact, we are currently investigating this in detail, since we are developing a new vaccine for rheumatoid arthritis based on the knowledge of such mechanisms.

R2-Q5: *Finally, the transfer of bulk DCs isolation from WT mice or MMC mice leads to the changes in HCQ3 SP CD4 T-cells frequency (Fig. 5f). However, in this particular setting, there is no direct evidence provided, that DCs are able to bring Col2a1 as the antigen directly into the thymus. The observed change in T-cell frequency can be viewed as the consequence of immune reaction in the periphery or perturbation in the immune system after the transfer. Probably, the intrathymic injection of the WT and PTM peptide should be tried. Alternatively, for instance CFSE-labeled DCs can be pulsed with WT or PTM peptide and transfer to HCQ3 mice. The migration of DCs into the thymus can be monitored, as well as their impact on HCQ3 T-cells selection.*

The transfer of DC from WT or MMC mice as a possible mechanism of inducing thymic negative selection was based on the fact that adoptively transferred HCQ3 cells proliferate in lymph nodes from MMC mice but not WT (supplementary figure 6). This indicates that the cognate antigen (PTM CII) is present in secondary lymphoid organs of MMC mice but not WT mice. We isolated DCs from these mice based on their pivotal role in inducing T cell activation in peripheral LN, as well as by their capacity of migrating to the thymus and promoting central selection (13, 14). A

similar experiment could be done by pulsing DCs with PTM CII₂₆₀₋₂₇₀ prior to transfer. However, we considered that transferring untouched DCs would resemble a more natural way of central tolerance to occur. A “forced” mechanism such as DC pulsing or thymic injection of the peptide would very possibly result in higher frequency of cognate peptide-MHC complexes available for negative selection, and therefore confer a bias in the selection process. Moreover, simple manipulation of DCs *ex vivo*, such as antigen pulsing, will activate DCs and disturb their migration capacity to the thymus (14). On the other hand, direct injection of the PTM peptide into the thymus will, expectably, result in negative selection of HCQ3 cells. However, it will not elucidate by which way such selection could occur *in vivo*. Nevertheless, in the same line of thought, we have previously shown that neonatal injection of PTM CII₂₆₀₋₂₇₀ peptide, but not native peptide, is able to induce arthritis protection later in adulthood (3). This observation further supports the hypothesis that periphery-derived PTM CII is of relevance to induce T cell selection in the thymus, and that this process is of importance already at neonatal stage.

Minor issues:

R2-q1. *In general, I would suggest using scatter plots instead of bar graphs in order to visualize better values distribution.*

We considered that only figure 2 would benefit from having each individual point being represented rather than a simple bar chart. Hence, all representations remain the same for the exception of figures 2b and 2c.

R2-q2. *Since fibroblasts can also express different types of collagens, it would be beneficial to determine if mTECs are the only source of Col2a1 expression within the thymus by FACS-sorting of thymic fibroblasts, mTEC_{slow}, mTEC_{high}, and possibly other thymic populations and subsequent qPCR analysis.*

Fibroblasts produce collagens but are not known to produce type II collagen. Nevertheless, in artificial situations (lentivirus transfection) fibroblasts can produce and even glycosylate CII, as discussed above. However, even if other cells than mTECs would express CII, we did not detect it through functional analysis using Aire^{KO} mice, or via direct staining of mouse and human thymus. In that regard, the newly added data, where the stain with T8 clone (PTM-specific) was compared to that of other anti-CII mAb, clearly confines CII to medullary cells. In nasal and joint cartilage, CII and glycosylated CII can be detected in cartilage and chondrocytes, but not in other cells.

R2-q3. *In page 6 line 122 “differentially” should be changed for “diferentially”.*

The word has been corrected.

R2-q4. *Term “data not shown” (Line 147 page 7 & line 226 page 10) should be excluded, instead data should be provided as supplementary information.*

The term has been removed from line 147. The data refers to the confirmation of nucleotide sequence amplified by PCR. It is enough to state this, since the sequence was as expected and, thus, there are no details needed to be shown as supplementary data.

R2-q5. *There is a lack of clarity in lines 155-160. It is not clear enough how this system would allow to know if negative selection of CII specific T cells results from the presentation of the non-modified CII or an unrelated auto-antigen. The authors might consider rewriting those sentences.*

We have now rephrased this section, which is important to connect the results shown before (figures 1, 2 and 3) and the subsequently performed experiments. In other words, the lines 156-160 (now 177-181) hypothesize a scenario where the data presented in figures 1, 2 and 3 would be the result of the controlled expression/presentation of an unrelated cross-reactive antigen. In order to exclude this possibility, and confirm that those observations result from a direct influence of central mechanisms of CII expression/presentation, we performed the experiments shown in figure 4.

R2-q6. *The statement Col2a1mRNA transcript was detected only in the thymus but not in other lymphoid organs is rather inaccurate (pg. 6-7, 145-147). As the corresponding figure shows only the absence of Col2a1mRNA transcript in the spleen. The statement should be rephrased or PCR detection of Col2a1 expression in other lymphoid organs should be included.*

The statement has been rephrased, where the word “spleen” has been included.

R2-q7. *Mice carrying the Ncf1 mutation are well described in methods section but they are not even mentioned in the results section, however the Fig.1 is completely based on them. Please include the rationale of using this particular strain directly into the result section.*

Due to word count restrictions, the description of the Ncf1 mutated animals was kept in the *Methods* section. We have now inserted a sentence explaining why the Ncf1 mutation was used; “All mice had a mutation in the *Ncf1* gene, which enhances arthritis susceptibility and allows for the development of arthritis in MMC mice (Hultqvist M. et al. 2007, J. Immunol.)”

R2-q8. *Fig.1 – Ncf vs. Ncf1. The strain name should be used consistently in graph legends.*

All graphs are now consistent, stating Ncf1.

R2-q9. *In line 110-112 (referring to Fig 1b) authors state that in MMC mice response to the native form is almost abrogated in comparison with the PTM form. What was the statistical test applied? This information should be included. If what was tested is MMC mice response to native vs. PTM, Wilcoxon should be used instead of Mann Whitney.*

We agree that it is better to use the Wilcoxon paired sample test rather than the Wilcoxon ranked sum test (also called Mann Whitney test) in this case, since the analysis concerns paired samples. We have now calculated the statistics between the native and PTM response in BQ.Ncf1.MMC (figure 1b) using the Wilcoxon paired sample test instead of Mann Whitney as initially performed: $p = 0.0195$ for a 95% confidence interval. The PTM response in these mice is significantly higher than the native form. The median number of spots observed in response to the native peptide was 2.3, whereas the median to PTM peptide was 14.3 spots.

R2-q10. *To support that MMC mice are less susceptible to arthritis (Fig1c) it would be valuable to include the histology of the cartilage or pictures of arthritic limbs showing the severity of the lesions. (This also applies to Fig 2a)*

The concept of “arthritis susceptibility” regards to the prevalence of disease observed between the different mouse strains, and not to the severity of the disease. In figure 1c and figure 2a we show disease prevalence of wild type versus MMC and Aire^{Suf}.MMC versus Aire^{KO}.MMC, respectively. In each case, the susceptibility of developing arthritis after disease induction is significantly higher in wild type and Aire^{KO}.MMC. We do not make any claims in terms of disease severity, since those (few) mice that develop disease will manifest similar arthritic symptoms as their littermates. In fact, this observation supports our hypothesis that peripheral tolerance mechanisms are likely to be insufficient in conferring tolerance to PTM self-antigens.

R2-q11. *In line 131 “mice mounted a significant response only...” was there a statistical evaluation performed? If not, the word “significant” is not justified.*

To avoid confusion with the terminology used, “significant” is now replaced by “substantial”.

R2-q12. *In figure legend 3a, the authors do not provide the information about the displayed data. Information regarding whether the bars show SD or SEM etc. should be provided.*

The information has been added to the figure legend. SEM is being displayed.

R2-q13. *Fig. 3 and 5 are rather blurry and their resolution is low.*

We apologize for this. Higher resolution images will be uploaded in the revised version of the manuscript. Figure 3c has been moved to supplementary figure 4., whereas new imaging data regarding identification of PTM CII has been included in figure 3.

R2-q14. *Immunofluorescence images (Fig 3 and Supplementary figure 4) should be stained with Aire, CII and some cytoplasmic or cell surface marker in order to correlate Aire and CII protein presence in mTEC and to verify the localization of CII within mTECs.*

New images have been added to the original data in order to generate a quantitative assessment of CII⁺ mTECs (figure 3, supplementary figure 4 and supplementary table 1). These images clearly show that CII is present in a cytoplasmic location. The anti-CII mAb used are conformation restricted and can therefore only recognize triple helical structures of CII. Thus being, these mAb will not detect linear sequences in e.g. peptide-MHC complexes. Moreover, keratin stain has been added to some of the new panels of figure 3.

R2-q15. *In figure 5 a and b authors show reduction in the frequency of CD4 and CD40L+ cells in HCQ.MMC mice in comparison with HCQ mice. It would be valuable to add cell counts as well as to check if Treg compartment is affected.*

The data in figure 5a and 5b regards to *in vitro* stimulation of cells, where a given number of cells was used per cell culture well. Conversion of the frequency of cells into whole organ count is thus inaccurate, and therefore not presented. As mentioned

above, the analysis of the Treg compartment is outside the scope of this manuscript. Nevertheless, the precise role of Tregs is an important but complex issue and cannot be solved easily. We are currently working in these issues as also explained above.

R2-q16. *Instead of no antigen control the irrelevant peptide control (e.g. OVA) should be used in antigen proliferation or activation experiments using CII specific HCQ3 T cells (Fig 5).*

The HCQ3 TCR transgenic mouse was constructed using the HCQ3 T cell hybridoma as a reference (4). The specificity of HCQ3 cells to the PTM CII peptide has been previously demonstrated (3, 4, 7, 15). Moreover, we show in figure 1d that HCQ3 cells do not respond to native (not posttranslational modified) CII.

R2-q17. *The Suppl. Fig. 6 presents proliferation of HCQ3 cells in MMC mice after their transfer. Are HCQ3 T-cells itself able to induce arthritis in MMC mice? If not, is the transfer of HCQ3 cells to T-cell deficient animal model crossed with MMC mice able to promote it?*

Transfer of HCQ3 cells to MMC does not result in arthritis. The tolerance mechanisms involved are likely to be the same as those in MMC mice immunized with heterologous CII. The capacity of transferred HCQ3 T cells to elicit arthritic symptoms in a T cell-deficient MMC recipient is a very interesting question, and is being included in our current work to more precisely understand the role of Tregs in mediating tolerance in the MMC mouse.

References

1. Michaëlsson E et al. *The Journal of Experimental Medicine*. 180(2):745. (1994)
2. Michaëlsson E et al. *European Journal of Immunology*. 26(8):1906. (1996)
3. Bäcklund J et al. *European Journal of Immunology*. 32(12):3776. (2002)
4. Corthay A et al. *European Journal of Immunology*. 28(8):2580. (1998)
5. Broddefalk J et al. *J. Am. Chem. Soc.* 120:7676. (1998)
6. Holm B et al. *Chembiochem*. 3(12):1209. (2002)
7. Dzhambazov B et al. *European Journal of Immunology*. 35(2):357. (2005)
8. Gjertsson I et al. *Mol. Ther.* 17(4):632. (2009)
9. Raposo B et al. *Journal of Experimental Medicine*. 20(15):3031. (2014)
10. Malmström V et al. *Arthritis Research*. 2(4):315. (2000)
11. Treschow AP, Bäcklund J, Holmdahl R, Issazadeh-Navikas S. *J Immunol*. 174(11):6742. (2005)
12. Bäcklund J et al. *J Immunol*. 171(7):3493. (2003)
13. Hadeiba H et al. *Immunity*. 36(3):438. (2012)
14. Bonasio R et al. *Nature Immunology*. 7(10):1092. (2006)
15. Dzhambazov B et al. *J Immunol*. 176(3):1525. (2006)

REVIEWERS' COMMENTS:

Reviewer #2 (Remarks to the Author):

Although the authors have improved the original manuscript, there are two major issues that preclude publication of this ms in the current form (in the opinion of this referee).

1) I am sorry to be repetitive, but one cannot make general conclusions that are based on a single case example. Such generalized, yet generally unsupported conclusions are very misleading, unfair and dangerous.

Therefore, if the authors are not able to provide sufficient experimental evidence supporting their general conclusion that "T cells specific for posttranslational modifications escape intrathymic tolerance induction", then they should at least be more specific and less general. The only conclusion one may derive from this ms is that tolerance to post-translationally modified collagen doesn't seem to be induced by bmTEC/Aire -dependent mechanisms. Thus, the title and the abstract should reflect concrete results that are supported experimentally, rather than overgeneralized statements, based on pure speculations!

The authors may speculate in the discussion that this MAY be a more general phenomenon, but they should not do so in the title/abstract/results.

The results do not have to be over-interpreted to still have an important scientific value

2) The mechanisms underlying this phenomenon are not well delineated. Is the clonal deletion or agonist selection of Tregs in the thymus the dominant mechanism? The evidence that the tolerance is mediated in the thymus by DCs importing the PTM-Ag from the periphery is rather weak.

Point-to-point response to reviewer 2 comments on the revised manuscript by Raposo et al.

Reviewer's comment in italic:

Although the authors have improved the original manuscript, there are two major issues that preclude publication of this ms in the current form (in the opinion of this referee).

Point 1) *I am sorry to be repetitive, but one cannot make general conclusions that are based on a single case example. Such generalized, yet generally unsupported conclusions are very misleading, unfair and dangerous.*

Therefore, if the authors are not able to provide sufficient experimental evidence supporting their general conclusion that "T cells specific for posttranslational modifications escape intrathymic tolerance induction", then they should at least be more specific and less general. The only conclusion one may derive from this ms is that tolerance to post-translationally modified collagen doesn't seem to be induced by mTEC/Aire-dependent mechanisms. Thus, the title and the abstract should reflect concrete results that are supported experimentally, rather than overgeneralized statements, based on pure speculations!

The authors may speculate in the discussion that this MAY be a more general phenomenon, but they should not do so in the title/abstract/results.

The results do not have to be over-interpreted to still have an important scientific value

Reply point 1: As described in the point-to-point reply of the first revision of the manuscript it is not feasible at this point to set up an additional model to confirm our current observations as the corresponding models and tools are not available. However, we do agree with reviewer 2 that one cannot claim at this point that our observations on the escape from tolerance to PTM antigen constitute a general mechanism whereby T cells evade tolerance. We therefore emphasize this even further in the second revision of the manuscript (in the abstract as well as in the beginning and end of the discussion). Furthermore, we clearly emphasize that the current study only involves tolerance to CII.

Point 2) *The mechanisms underlying this phenomenon are not well delineated. Is the clonal deletion or agonist selection of Tregs in the thymus the dominant mechanism? The evidence that the tolerance is mediated in the thymus by DCs importing the PTM-Ag from the periphery is rather weak.*

Reply point 2: One of the major findings of the current study is that T cell tolerance to the PTM epitope is relatively weaker when compared to that induced by the native epitope. This is true, despite that the PTM variant is present in both the periphery and (to some extent) in the thymus, whereas the native variant appears only to be present in the thymus. However, it is not clear at this point whether tolerance to the PTM version of CII observed in MMC mice occurs preferentially in the periphery, by presentation of the PTM epitope in peripheral lymphoid organs, or in the thymus via the transport of the antigen by

migratory DC. This is a very interesting question that is now possible to start addressing after the present observations and using the model systems that have been established for the current manuscript. Still, the exact nature of tolerance is likely to be very complex and therefore out of the scope of the current report.